# Monitoring site-specific conformational changes in real-time reveals a misfolding mechanism of the prion protein

Ishita Sengupta†*, Jayant Udgaonkar‡*

National Centre for Biological Sciences, Tata Institute of Fundamental Research, Bengaluru, India

**Abstract** During pathological aggregation, proteins undergo remarkable conformational re-arrangements to anomalously assemble into a heterogeneous collection of misfolded multimers, ranging from soluble oligomers to insoluble amyloid fibrils. Inspired by fluorescence resonance energy transfer (FRET) measurements of protein folding, an experimental strategy to study site-specific misfolding kinetics during aggregation, by effectively suppressing contributions from inter-molecular FRET, is described. Specifically, the kinetics of conformational changes across different secondary and tertiary structural segments of the mouse prion protein (moPrP) were monitored independently, after the monomeric units transformed into large oligomers $O_L$, which subsequently disaggregated reversibly into small oligomers $O_S$ at pH 4. The sequence segments spanning helices $\alpha2$ and $\alpha3$ underwent a compaction during the formation of $O_L$ and elongation into β-sheets during the formation of $O_S$. The β1-α1-β2 and α2-α3 subdomains were separated, and the helix α1 was unfolded to varying extents in both $O_L$ and $O_S$.

DOI: https://doi.org/10.7554/eLife.44698.001

*For correspondence:
ishita@iitk.ac.in (IS);
jayant@iiserpune.ac.in (JU)

Present address: †Department of Chemistry, IIT Kanpur, Kanpur, India; ‡Department of Biology, Indian Institute of Science Education and Research (IISER) Pune, Pune, India

Competing interests: The authors declare that no competing interests exist.

## Introduction

The structural characterization of kinetic intermediates in protein aggregation is a challenging task. Most experimental probes, used to study misfolding and aggregation kinetics, track either the acquisition of β-structure, or global changes in size. Although intermediate forms populated transiently during fibril formation reactions can be detected, for example by single-molecule FRET measurements (*Cremades et al., 2012*; *Orte et al., 2008*; *Shammas et al., 2015*; *Yang et al., 2018*), their detailed structural characterization is difficult. Equilibrium and kinetic measurements using multi-site FRET, to probe conformational changes in different parts of a protein, while it folds, unfolds, forms functional oligomers or interacts with its binding partner, have been a rich source of site-specific information, usually invisible to global probes (*Lakshmikanth et al., 2001*; *Lillo et al., 1997*; *Lin et al., 2013*). Using a similar approach to study misfolding can potentially reveal the step-wise conformational changes that lead to the global misfolding of a protein.

The mostly α-helical and monomeric prion protein (PrP) undergoes drastic secondary and tertiary structural re-arrangements upon aggregation into a variety of misfolded β-sheet-rich multimers (*Pan et al., 1993*), most of which are not infectious. A class of fatal neurodegenerative diseases collectively known as transmissible spongiform encephalopathies are caused by infectious aggregates of misfolded PrP. Conversion to the pathogenic form possibly initiates in the endocytic pathway, when the protein encounters an acidic environment (*Borchelt et al., 1992*). The pathogenic misfolded aggregates thus formed in vivo are highly heterogeneous, with the most infectious oligomers composed of 14–28 monomers (*Silveira et al., 2005*).

The extremely rugged aggregation landscape of PrP has made it challenging to determine the high-resolution structure(s) of its various misfolded β-rich aggregated forms. However, several

structural models have been derived from experimental data, which differ not only in their secondary structure content, but also in the location and size of the β-sheet-rich core. While some models suggest that a major part of the native fold remains intact in the aggregates (*DeMarco and Daggett, 2004*; *Govaerts et al., 2004*), several experimental studies have shown that a major section of the β-rich core of the aggregates formed from the full length PrP is located in the sequence segment that corresponds to the α2-α3 subdomain of monomeric PrP (*Cobb et al., 2007*; *Diaz-Espinoza and Soto, 2012*; *Singh et al., 2012*; *Tycko et al., 2010*). It has been a challenge to determine the mechanism of misfolding that occurs during the formation of any of the distinct β-rich aggregates of PrP.

In agreement with observations made in vivo, the oligomerization and misfolding of mouse PrP (moPrP) is favoured in vitro at low pH in the presence of 150 mM NaCl (*Singh et al., 2014*; *Singh and Udgaonkar, 2015b*), due to the protonation of residues H186 and/or D201 in the α2-α3 subdomain of moPrP. Previous HX-MS studies (*Singh et al., 2012*; *Singh and Udgaonkar, 2015a*) had shown that the protected core region of the β-rich oligomers formed by PrP at low pH in vitro resembles that of amyloid fibrils derived from diseased brain. Not surprisingly then, recombinant PrP from animal species with high susceptibility to prion disease has been shown to readily form β-rich oligomers at low pH in vitro (*Khan et al., 2010*). β-rich oligomers formed at low pH, readily disrupt lipid membranes; this property is a likely reason for their toxicity (*Singh et al., 2014*; *Singh et al., 2012*). Thus, the β-rich oligomers formed at low pH appear to be a suitable structural model for studying a putatively important misfolding mechanism of PrP.

Interestingly, oligomer formation of moPrP at pH 4 could be completely abolished by substituting a discordant but highly conserved sequence stretch (TVTTTT) in the C-terminal end of α2, with a very high β-sheet propensity (*Dima and Thirumalai, 2002*), by the α-helix favouring residue alanine (AAAAAA) (*Singh et al., 2014*). This suggests that the C-terminal end of α2 plays a critical role in the initiation of misfolding. In species with low susceptibility to prion disease, the loop between β2 and α2 is more rigid in the monomeric prion protein (*Gossert et al., 2005*), suggesting that its flexibility might play an important role in facilitating misfolding. It is also known that residues H186 and D201 together with R155, K193 and E195 form a network of electrostatic interactions between the α2-α3 and β1-α1-β2 subdomains in monomeric PrP (*Hadži et al., 2015*; *Hosszu et al., 2010*; *Singh and Udgaonkar, 2015b*). The disruption of these electrostatic interactions either by a lowering of pH (*Singh and Udgaonkar, 2016a*) and addition of salt (*Sengupta et al., 2017*), or by charge reversing or neutralizing pathogenic mutations (*Singh and Udgaonkar, 2016a*; *Singh and Udgaonkar, 2015a*) facilitate misfolding and oligomer formation in vitro. It appears that separation of the α2-α3 and β1-α1-β2 subdomains must occur before conversion to the β-conformation (*Eghiaian et al., 2007*; *Hafner-Bratkovic et al., 2011*). Locking of the α2-α3 and β1-α1-β2 subdomains either by engineering in an artificial disulphide bond (*Eghiaian et al., 2007*), or by binding to anti-prion drugs (*Kamatari et al., 2013*) prevent misfolding and aggregation. While such equilibrium studies have suggested possible sequences of structural changes during oligomer formation (*Singh and Udgaonkar, 2016a*), there has been a dire need for kinetic studies that can directly delineate the structural mechanism of the misfolding which accompanies the oligomerization of the prion protein at low pH.

In the current study, the structural mechanism of the spontaneous formation of β-rich oligomers at pH 4, in the absence of any denaturant but in the presence of 150 mM NaCl, has been delineated. Earlier studies of the formation of these misfolded oligomers had indicated that oligomerization drives major intra-molecular conformational change. Real-time NMR measurements had shown that no major conformational change occurs in the monomer before it assembles into oligomer: the monomer structure is perturbed to only a very minor extent before assembly into oligomers, which is rate-limited by association steps in which dimers and trimers are formed (*Sengupta et al., 2017*). Not surprisingly, single-molecule studies have found that misfolded monomeric PrP is not stable (*Yu et al., 2012*). Measurements of changes in size and conformation by CD (circular dichroism), SEC (size-exclusion chromatography) and HX-MS (hydrogen-exchange coupled to mass spectrometry), had shown that oligomerization is faster than major conformational change in the case of two pathogenic mutant variants of moPrP (*Sabareesan and Udgaonkar, 2016*), under oligomerization conditions identical to those used in the current study. Notably, β-rich oligomers of moPrP, which form worm-like fibrils by an isodesmic mechanism at pH 2 (*Jain and Udgaonkar, 2008*), are distinct from the off-pathway oligomers seen to form transiently during nucleation-dependent amyloid fibril formation (*Sabareesan and Udgaonkar, 2017*), from the octamers formed at low pH in the presence

of 2 M urea (*Larda et al., 2013*), from the oligomers formed spontaneously by a tandem dimer (*Gilch et al., 2003*), and from the neurotoxic oligomers obtained by thermal refolding of ovine prion protein (*Eghiaian et al., 2007*; *Rezaei et al., 2005*).

Most kinetic studies of protein misfolding have lacked sufficient structural resolution or have been complicated by the effects of multimerization. Deriving inspiration from site-directed spin labelling (SDSL) EPR and FRET measurements on protein aggregates (*Margittai and Langen, 2008*) and very successful kinetic studies of protein folding, using multi-site FRET (*Lakshmikanth et al., 2001*; *Lillo et al., 1997*) a generally applicable method was developed to measure site-specific misfolding kinetics in non-nucleated aggregating systems, while eliminating the complicating effects of multi-merization. This was achieved using FRET between a Trp residue and a thio-nitrobenzoate (TNB)-moiety attached to a free Cys residue (*Lakshmikanth et al., 2001*), in five different single Trp, single Cys-containing mutant variants of moPrP. Every pair of proteins, with and without the TNB adduct, was separately co-oligomerized with a large excess of a tryptophan-free (Trp-less) mutant variant of moPrP. Such an experimental strategy has been shown to result in the suppression of inter-molecular contributions to FRET (*Duim et al., 2014*; *Pinotsi et al., 2014*); hence, the FRET measurements report only on structural changes occurring in each monomeric unit comprising the oligomer.

To complement the FRET studies, which were used to monitor conformational changes within each monomeric unit of the oligomer, techniques that monitored changes in the global properties of the oligomers were employed. CD was used to monitor global changes in secondary structure (mostly β-sheet formation), and steady-state tryptophan fluorescence intensity and anisotropy were used to measure changes in the local environments of differently placed Trp residues, and in overall oligomer size/population, respectively. Size-exclusion chromatography was used to monitor the kinetics of monomer loss during oligomer formation, and to characterize the heterogeneity in the oligomer formation. It should be noted that by themselves probes such as CD, cannot detect site-specific structural changes, because the signal is dominated by β-sheet formation, making it insensitive to any other conformational change. In contrast, the experimental strategy demonstrated here, allows the visualization of segment-specific misfolding of moPrP, during the course of oligomer formation at low pH. It is, however, not possible to distinguish whether all monomeric units undergo these conformational changes synchronously or in a random manner, using this approach.

Taken all together, the fluorescence and FRET data revealed that a local perturbation in the loop separating $\alpha2$ and $\alpha3$ took place prior to oligomer formation. Along the course of the oligomerization reaction, the fastest change appeared to be the compaction of the sequence segments spanning helices $\alpha2$ and $\alpha3$. The separation of the $\alpha2$-$\alpha3$ sub-domain from the $\beta1$-$\alpha1$-$\beta2$ subdomain appeared to be slower. The slowest changes appeared to be the unfolding of $\alpha1$, and the expansion of the sequence segments that encompassed $\alpha2$ and $\alpha3$ into extended β-strands. From the size-exclusion chromatography results, two oligomeric species of distinct sizes, $O_L$ and $O_S$, were seen to be populated to varying extents at all times of the oligomerization reaction, suggesting that monomer M, $O_L$ and $O_S$ were interconverting. Kinetic modelling and global fitting of the conversion of M to $O_L$ and $O_S$, revealed that the contraction of the sequence segments spanning helices $\alpha2$ and $\alpha3$ took place concomitant to the formation of oligomer $O_L$ from monomers, and their expansion took place as $O_L$ disaggregated reversibly to form $O_S$. $\alpha1$ was unfolded and the $\alpha2$-$\alpha3$ sub-domain was separated from the $\beta1$-$\alpha1$-$\beta2$ subdomain to varying extents in both $O_L$ and $O_S$.

## Results

### FRET pairs to monitor site-specific misfolding in the monomeric unit of the oligomer

The Trp-TNB FRET pair has an estimated Forster radius ($R_0$) of ~23 Å (*Supplementary file 1*), which allows the reliable measurement of change in distance in the range of ~11 to ~35 Å (*Lakshmikanth et al., 2001*). WT moPrP has eight Trp residues, seven of which are located in the intrinsically disordered N terminal region (NTR), and one (W144) at the N terminus of $\alpha1$ in the structured C-terminal domain (CTD). All native Trp residues are solvent-exposed in the WT moPrP monomer. The NTR remains disordered in the oligomer, and major structural changes take place in the CTD upon oligomerization (*Sengupta et al., 2017*; *Singh and Udgaonkar, 2015a*). To monitor these changes, five single Trp, single Cys-containing mutant variants in the CTD were designed for site-

specific FRET measurements: W144-C153 to monitor conformational changes in α1; W144-C199 and W144-C223 to monitor the separation of α1 from the α2-α3 subdomain; and W197-C169 and W197-C223 to monitor conformational changes within α2 and α3 (*Figure 1A*).

The donor fluorophore was W144 at the N terminus of α1 in three of the five mutant variants, and W197, in the loop between α2 and α3 in the other two variants (*Jenkins et al., 2008*). The native disulphide bond between residues C178 and C213 was intact in all of the constructs as judged by ESI-MS mass spectrometry (*Figure 1—figure supplement 1*). The extra Cys residue in each of the mutant variants was covalently modified with TNB (thionitrobenzoate), DNP-C2 (dinitrophenyl) (*Yu et al., 2012*) or DANS (5-(DimethylAmino)Naphthalene-1-Sulfonyl) to obtain the corresponding labelled protein(s). It is to be noted that it has been shown previously by HX-MS/NMR measurements, that α1 (which houses the buried C153 residue) is much more flexible than the hydrophobic

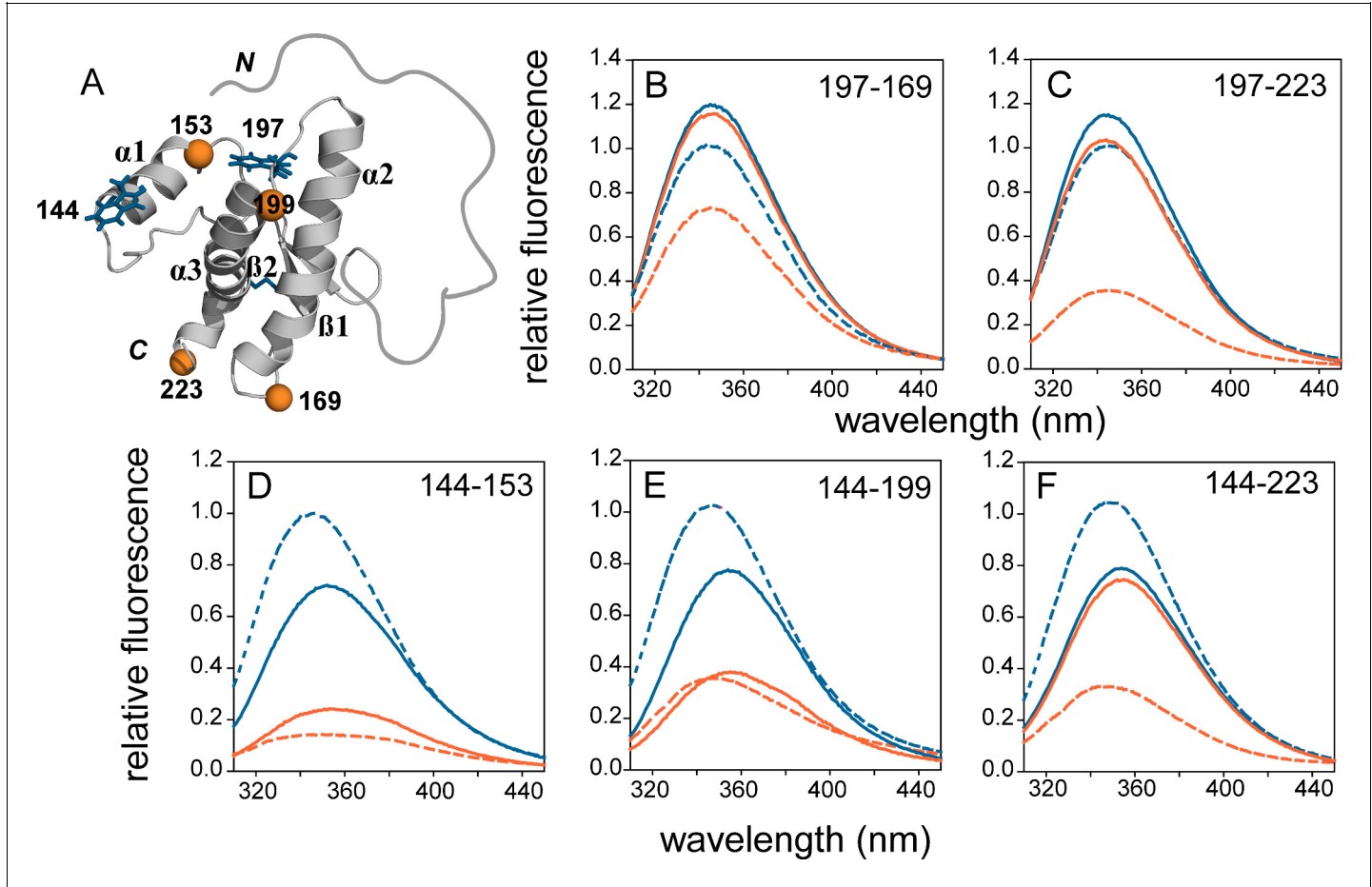

**Figure 1.** FRET in mutant variants of moPrP in their monomeric and oligomeric forms. (**A**) Structure of moPrP (PDB ID 1AG2) showing the positions of the FRET pairs. The donor tryptophans, W144 and W197 are shown as blue sticks, and the acceptor cysteines with covalently bound TNB moieties are shown as orange spheres. The five single Trp, single Cys-containing mutant variants, corresponding to W144-C153, W144-C199, W144-C223, W197-C169 and W197-C223 are shown. The secondary structural elements, N and C-termini and disulphide bond are indicated. (**B-F**) Fluorescence emission spectra of unlabelled (blue) and TNB-labelled (orange) the single Trp, single Cys-containing mutant variants in their monomeric (solid lines) and oligomeric forms (dashed lines) are shown. The relevant FRET pairs are indicated in each panel.

DOI: https://doi.org/10.7554/eLife.44698.002

The following figure supplements are available for figure 1:

**Figure supplement 1.** ESI-MS spectra of all unlabelled and labelled mutant variants.

DOI: https://doi.org/10.7554/eLife.44698.003

**Figure supplement 2.** Thermodynamic stability and far-UV CD monitored misfolding kinetics of Trp-less and labelled and unlabelled single Trp, single Cys-containing mutant variants of moPrP.

DOI: https://doi.org/10.7554/eLife.44698.004

core of the protein which houses the disulfide bond. In fact, residues in the hydrophobic core do not exchange even after 40 days under these conditions (*Moulick et al., 2015*). Therefore, while the fast dynamics of α1 allow the buried C153 side-chain to get labelled, the disulfide bond between residues C178 and C213 remains oxidized in that timescale. A Trp-less mutant variant, with all Trp residues mutated to Phe residues was also purified for co-oligomerization experiments, in which it could be shown that inter-molecular FRET was suppressed during oligomerization.

## Fluorescence and FRET efficiency changes in monomeric and oligomeric moPrP

Prior to the kinetic experiments, fluorescence emission spectra were recorded for each pair of unlabelled and TNB-labelled proteins, in their monomeric and oligomeric forms. The fluorescence emission maximum was at 355 nm and 345 nm, for monomeric mutant variants with W144 and W197 as the donor fluorophore, respectively (*Figure 1B–F*). This observation, indicating a markedly different local environment around each Trp residue, is in agreement with the solution NMR structure of monomeric moPrP, where W144 is completely exposed to solvent, and W197 is partially buried (*Riek et al., 1996*). In the corresponding TNB-labelled monomeric mutant variants, the Trp fluorescence was quenched, but to different extents. The extent of quenching by TNB was dependent on the separation of the Trp and the TNB moiety in the monomer, clearly indicative of FRET. In addition, the observed FRET efficiencies for different pairs of mutant variants were in good agreement with the expected FRET efficiencies calculated from the solution NMR structure (*Figure 1A*). The small differences in the expected and observed FRET efficiencies can be attributed to the size and orientation of the Trp and TNB-side chains (*Supplementary file 1*). Moreover, the far-UV CD spectra of all unlabelled and TNB-labelled mutant variants matched well with that of WT moPrP (*Figure 1—figure supplement 2*), indicating that the secondary structure is conserved in these proteins.

While all the unlabelled and TNB-labelled mutant variants were able to form β-rich oligomers, comparable to those formed by WT moPrP as judged by CD and DLS measurements (*Sengupta and Udgaonkar, 2017*), (*Figure 1—figure supplement 2*), the local environment around donor fluorophores W144 and W197 changed in different ways upon oligomer formation. A blue shift to 345 nm and an increase in quantum yield was observed for the W144-containing mutant variants, suggesting that the local environment of W144 was more hydrophobic in the oligomers than in the monomer (*Figure 1D–F*). In contrast, a slightly decreased quantum yield without a change in the emission maximum was observed for the W197-containing mutant variants (*Figure 1B–C*) indicating that the W197 side-chain was partially buried in both monomer and oligomer, but that additional quenching mechanisms were operative in the oligomer. The W197 side-chain in the monomer is in close proximity to residues H186 and Y155, which like the disulfide bond, can quench Trp fluorescence, either by excited state proton and electron transfer, or by direct contact, respectively (*Hennecke et al., 1997*; *Lakowicz, 2006*). Rearrangements in structure in each monomeric unit of the oligomer, or between monomeric units within the oligomer, can result in quenching, due to the proximity of these residues to W197. The enhanced FRET efficiency in the oligomers compared to the monomers, in all five mutant variants, suggested that both intra- and inter-molecular FRET could be contributing to the quenching of the tryptophan fluorescence in the oligomers.

## Co-oligomerization with Trp-less moPrP results in the suppression of inter-molecular FRET

To suppress the inter-molecular contribution to FRET in the oligomers, and to exclusively measure intra-molecular FRET changes during oligomer formation, single Trp-containing labelled and unlabelled protein (dopant) were co-oligomerized with increasing amounts of Trp-less moPrP (while keeping the total protein concentration fixed at 100 μM) (*Toyama and Weissman, 2011*).

The FRET efficiencies were similar for the oligomers prepared at 1:34 (~3 mol%) and 1:50 (2 mol %) doping ratios (*Figure 2A*), and significantly lower than that determined for oligomers prepared from the dopant protein alone (i.e. without Trp-less moPrP), or for oligomers prepared at a 1:13 (~8 mol%) doping ratio. The data suggested that at doping ratios greater than ~1:30, inter-molecular FRET was effectively suppressed.

The moPrP oligomers formed at pH 4 are comparable to the β-sheet rich oligomers at pH 2 (*Jain and Udgaonkar, 2008*) (*Jain and Udgaonkar, 2010*) (*Jain and Udgaonkar, 2011*), with a

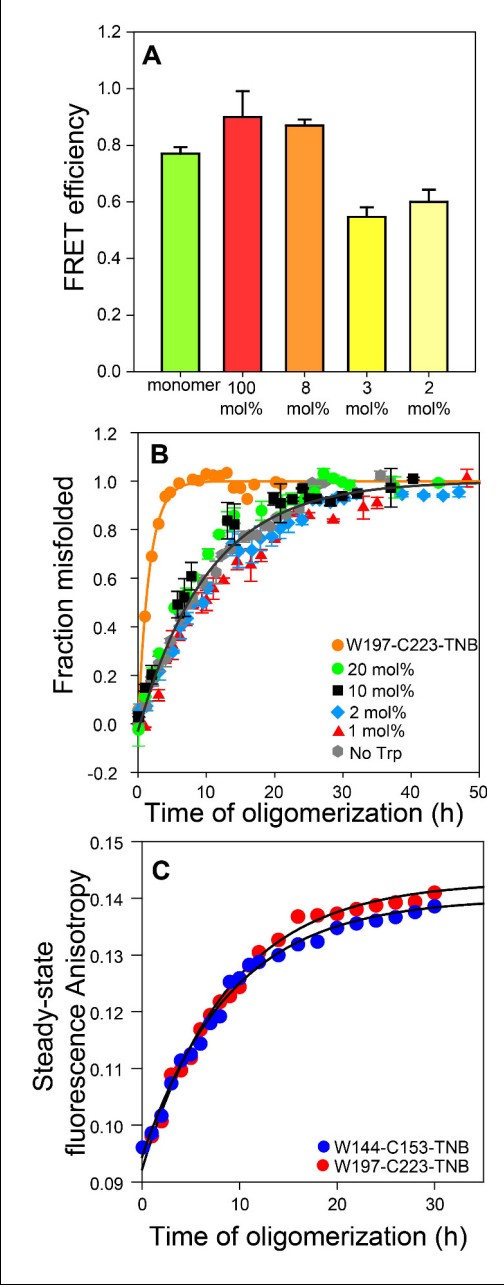

**Figure 2.** Co-oligomerization with Trp-less moPrP suppresses inter-molecular FRET but does not alter the global misfolding kinetics significantly. (**A**) Co-oligomerization with Trp-less moPrP suppresses inter-molecular FRET between monomers. Oligomers prepared from solely (100%) unlabelled and TNB-labelled W144-C153 moPrP (dopant) exhibit the highest FRET efficiency (red). To effectively suppress inter-molecular FRET between monomers that is units in the oligomer, oligomers were prepared in the presence of increasing concentrations of Trp-less moPrP, resulting in decreasing dopant concentrations of 8 (orange), 3 (bright yellow) and 2 mol% (pale yellow), while keeping the total protein concentration fixed at 100 µM. The FRET efficiencies were
*Figure 2 continued on next page*

molecular weight of 1207 ± 165 kDa from MALS measurements (*Singh et al., 2012*), corresponding to ~53 ± 7 monomeric units. Thus, a 1:50 doping ratio would correspond to one dopant molecule in every ~50 mer (oligomer), on the average. It was therefore not surprising that inter-molecular-FRET was effectively supressed at a 1:50 doping ratio.

## Unlabelled and TNB-labelled mutant variants form misfolded co-oligomers with Trp-less moPrP with comparable kinetics

An important pre-requisite for using FRET to monitor the kinetics of intra-molecular conformational change during oligomerization was that the different unlabelled and TNB-labelled mutant variants must co-oligomerize with Trp-less moPrP to form misfolded co-oligomers, with comparable rate constants. In order to verify that the fast misfolding kinetics of the dopant protein by itself had no influence on how fast it formed misfolded co-oligomers with Trp-less moPrP, the global misfolding kinetics of doped Trp-less moPrP was monitored by far-UV CD at four doping ratios (1:5, 1:10, 1:50 and 1:99) corresponding to 20, 10, 2 and 1 mol% of TNB-labelled W197-C223 moPrP (dopant) (*Figure 2B*). TNB-labelled W197-C223 moPrP was chosen as the dopant as it misfolds nearly 10-fold faster by itself at 100% labelling density, compared to Trp-less moPrP (*Supplementary file 2*). The global misfolding kinetics at 1:50 and 1:99 doping ratios were indistinguishable from that of only Trp-less moPrP. Only slightly faster kinetics was observed at 1:5 and 1:10 doping ratios (*Supplementary file 3*). It should be noted that although the observed kinetics appear to be described well by a single exponential equation, it is not possible to rule out the presence of two exponential components, with one component too small in amplitude to be detected.

The observations that at a 1:50 doping ratio, inter-molecular FRET was effectively suppressed in oligomers, and the CD-monitored kinetics of misfolding of Trp-less moPrP was unaffected, suggested that the use of this doping ratio was appropriate for monitoring the kinetics of intra-molecular FRET change during oligomerization.

To conclusively demonstrate that at a doping ratio of 1:50, the dopant protein and Trp-less moPrP did not oligomerize independently of each other, or that the oligomerization kinetics of Trp-less moPrP was altered, 2 µM dopant protein was mixed with 98 µM Trp-less protein, and the oligomerization reaction was monitored by

*Figure 2 continued*

comparable in the oligomers prepared from 3 and 2 mol% dopant concentrations, respectively, and significantly lower than that in the oligomers prepared from 100% dopant concentration. (**B**) Global misfolding kinetics of Trp-less moPrP, W197-C223-TNB and Trp-less moPrP doped with 1, 2, 10 and 20 mol% of W197-C223-TNB moPrP (dopant). Far-UV CD was the probe. The dopant by itself (orange) misfolds ~ 10 fold faster than Trp-less moPrP. The total protein concentration in each sample was 100 µM. Solid orange and gray lines represent the fit of the global misfolding data to single exponential kinetics for W197-C223-TNB and the Trp-less proteins, respectively. The misfolding rate constants are only marginally affected upon doping with increasing concentrations of the fast misfolding dopant. Error bars are standard deviation of the mean, determined from three independent measurements, from three separate samples. (**C**) Co-oligomerization of Trp-less moPrP with 2 mol% of W144-C153-TNB and W197-C223-TNB monitored by tryptophan steady-state fluorescence anisotropy.

DOI: https://doi.org/10.7554/eLife.44698.005

The following source data is available for figure 2:

**Source data 1.** Raw data for *Figure 2A–C*.
DOI: https://doi.org/10.7554/eLife.44698.006

steady-state Trp fluorescence anisotropy (*Figure 2C* and *Supplementary file 4*). This probe monitors the increase in size/population of oligomers that contain dopant protein (which contains the Trp fluorophore) and does not report on oligomers containing only Trp-less moPrP. Two dopant proteins, TNB-labelled W197-C223 and TNB-labelled W144-C153 were used at the 1:50 doping ratio. Although 100 µM TNB-labelled W197-C223 by itself misfolded more than fourfold faster than 100 µM TNB labelled W144-C153 (*Supplementary file 2*), the kinetics measured by steady-state fluorescence anisotropy was the same, no matter which of them was used as dopant at the 1:50 doping ratio. Moreover, the kinetics was comparable to the kinetics of CD-monitored misfolding of 100 µM Trp-less moPrP by itself. Taking the steady state fluorescence anisotropy measurements into consideration, it was not surprising to find that the CD-monitored misfolding kinetics of Trp-less moPrP doped with the different unlabelled and TNB-labelled mutant variants were similar (*Figure 3A*) at the

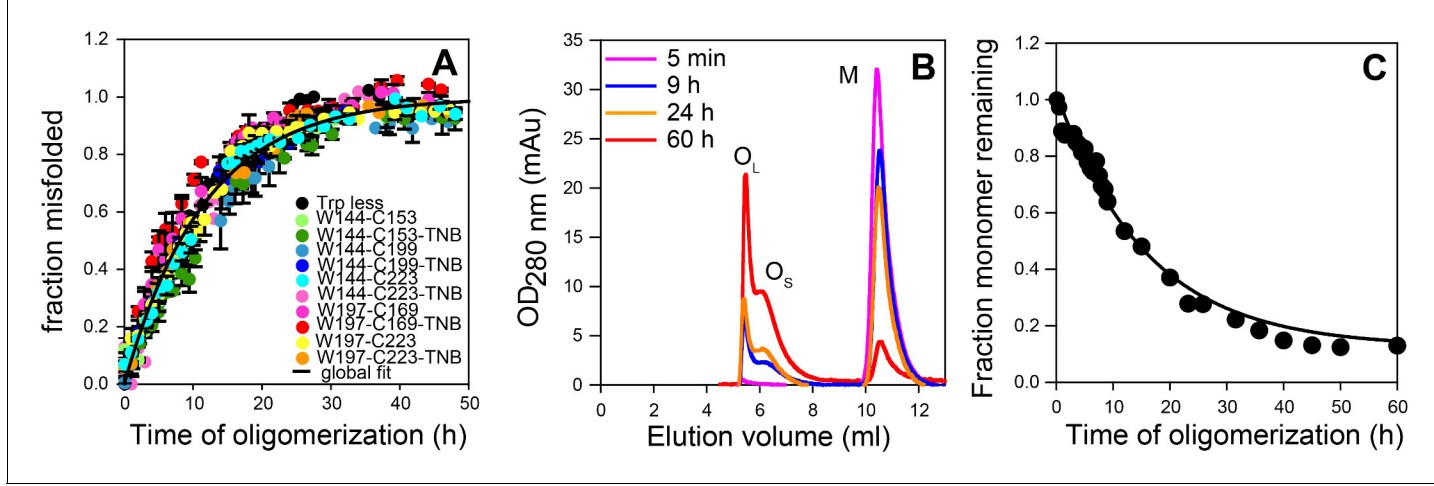

**Figure 3.** Global misfolding and oligomerization kinetics of Trp-less moPrP doped with 2 mol% of the unlabelled and TNB-labelled single Trp, single Cys-containing mutant variants of moPrP, monitored by CD and size-exclusion chromatography, respectively. (**A**) None of the dopant proteins altered the global misfolding kinetics of Trp-less moPrP to a significant extent at the dopant concentrations used in these experiments. The total protein concentration was kept fixed at 100 µM in all experiments. Global misfolding was monitored using far-UV CD. Error bars are standard deviation of the mean, determined from three independent measurements, from three separate samples. The global misfolding rate, as determined from a single exponential fit of all the data was 0.09 ± 0.03 h$^{-1}$. (**B**) Size-exclusion chromatograms at different time points of oligomerization of 100 µM Trp-less moPrP, showing the presence of oligomers $O_L$, $O_S$ and monomer M. (**C**) Normalized monomer loss kinetics, estimated from (**B**). The black line through the data is shown as a guide to the eye.

DOI: https://doi.org/10.7554/eLife.44698.007

The following source data and figure supplement are available for figure 3:

**Source data 1.** Raw data for *Figure 3*.
DOI: https://doi.org/10.7554/eLife.44698.009

**Figure supplement 1.** Trp-less moPrP shows a negligible change in fluorescence intensity upon oligomerization.
DOI: https://doi.org/10.7554/eLife.44698.008

1:50 doping ratio. Furthermore, at 2 µM concentration, either dopant protein by itself is expected to oligomerize with an apparent rate constant of $<0.02$ $h^{-1}$, from the known dependence of the oligomerization rate of moPrP on protein concentration (*Sabareesan and Udgaonkar, 2016*). The observation that the apparent rate constant from steady state Trp fluorescence anisotropy measurement was 0.1 $h^{-1}$, indicated that this probe reported on the co-oligomerization of dopant protein and Trp-less moPrp, and not on the independent oligomerization of 2 µM dopant protein.

The fluorescence spectra of the native monomers of the different unlabelled and labelled moPrP variants differed from those of the corresponding misfolded oligomers (*Figure 1*). Since there was no measurable change in fluorescence accompanying the misfolding and oligomerization of Trp-less moPrP by itself (*Figure 3—figure supplement 1*), the fluorescence change accompanying co-oligomerization of 2 µM dopant protein with 98 µM Trp-less moPrP, was easily measurable. Importantly, it was only when 2 µM dopant protein (any one of the unlabelled and TNB-labelled mutant variants) was mixed with 98 µM Trp-less moPrP, under oligomerization conditions, that a change in Trp fluorescence intensity was observed (*Figure 4*). 2 µM dopant protein alone (in the absence of Trp-less moPrP) under oligomerization conditions did not undergo any conformation-sensitive change in Trp fluorescence in ~30 hrs (*Figure 4*). These results also suggested that in the presence of 98 µM Trp-less moPrP, it was very unlikely that 2 µM dopant protein could misfold before oligomerization, and supported the results of the steady state Trp fluorescence anisotropy measurements which had indicated that the two proteins co-oligomerized.

Thus, when the dopant was present at 2 mol%, the kinetics of conformational change was not dependent on whether the dopant was labelled with TNB or not. Hence, the fluorescence-monitored kinetics of 2 mol% of the different unlabelled and TNB-labelled proteins could be compared directly in subsequent FRET measurements of site-specific misfolding.

## Oligomer formation monitored by size-exclusion chromatography

To directly monitor the change in population and/or size of the oligomers, size-exclusion chromatography (SEC) was carried out. In agreement with previous results obtained with WT moPrP (*Singh et al., 2014*), Trp-less moPrP was found to form two populations of oligomers, the larger $O_L$ and the smaller $O_S$, at pH 4 and 150 mM NaCl, As oligomerization progressed, the total amount of oligomers increased, but the sizes (as estimated from their elution volumes) remained fixed (*Figure 3B*). The kinetics of monomer loss/oligomer formation, as determined from the SEC measurements, were found to be apparently single exponential in nature, with a characteristic time (1/k) of $16.7 \pm 2.8$ hr (*Figure 3C*). Due to the limited resolution of the SEC column, it was not possible to separate the two oligomers and experimentally determine their FRET efficiencies. The populations of M and of $O_L$ and $O_S$ together were estimated to be ~14% and ~86%, respectively, after 60 hr of oligomerization. The poor resolution of the SEC column made it difficult to reliably estimate the relative amounts of $O_L$ and $O_S$ at equilibrium, but it appeared that, $O_S$ was populated to about a three-fold higher extent than was $O_L$.

## Local conformational change in moPrP monitored by fluorescence change

Changes in Trp fluorescence intensity can detect local conformational changes during co-oligomerization. Here, the co-oligomerization of the three unlabelled mutant variants containing W144 (W144-C153, W144-C199 and W144-C223) with Trp-less moPrP was accompanied by a slow ~80% increase in fluorescence in a single kinetic phase with characteristic times (1/k) of $8.3 \pm 1.3$, $6.7 \pm 2.7$ and $10 \pm 1$ hr, respectively (*Figure 4A–C* and *Table 1*). The amplitude of the signal change as well as the misfolding kinetics for all three unlabelled proteins was comparable. In marked contrast to the W144 variants, the two unlabelled mutant moPrP variants containing W197 (W197-C169 and W197-C223) underwent a quenching of fluorescence in two kinetic phases: a burst phase change (~40% amplitude) which was complete within the dead time of measurement (5 min), and a slow phase which accounted for the rest of the signal change (~60% amplitude) during the co-oligomerization reaction (*Figure 4D–E*). The characteristic times for the slow kinetic phase were $8.3 \pm 0.7$ and $8.3 \pm 2.1$ hr (*Table 1*). The characteristic times of the slow phase of fluorescence change for the different W144 and W197 mutant variants were comparable, and depended on the total protein concentration (while keeping the doping ratio fixed at 1:50), as expected for an assembly reaction

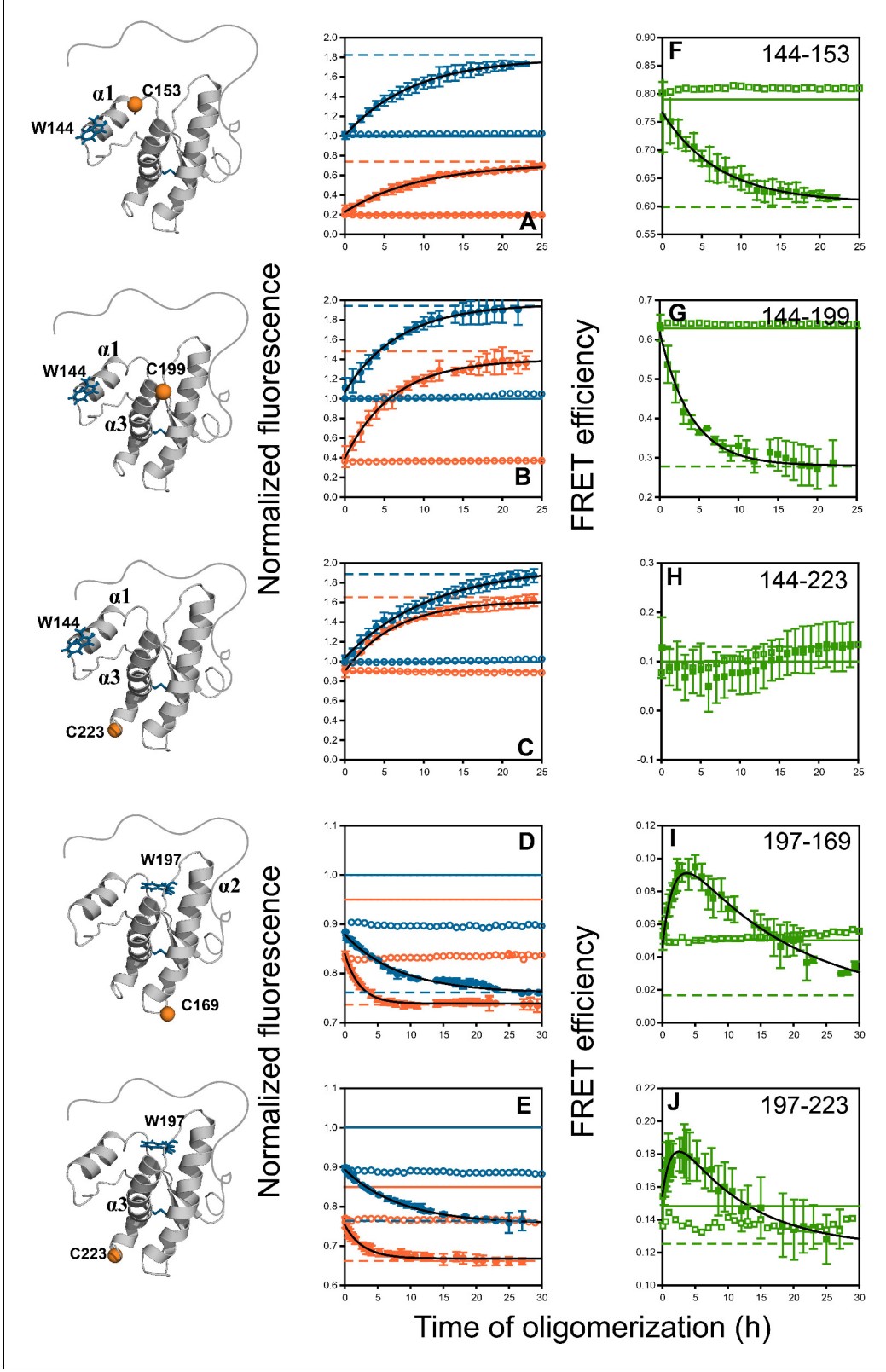

**Figure 4.** Monitoring misfolding by site-specific intra-molecular FRET. Unlabelled (blue) and TNB-labelled (orange) single Trp, single Cys-containing mutant variants W144-C153, W144-C199, W144-C223, W197-C169 and W197-C223 (A–E) were either co-oligomerized separately with Trp-less moPrP at a dopant concentration of 2 mol% (filled symbols), or in its absence (empty symbols). The corresponding changes in tryptophan fluorescence emission were measured as a function of time. The tryptophan fluorescence signal for the monomeric unlabelled (blue) and TNB-labelled (orange) protein(s) are

*Figure 4 continued on next page*

*Figure 4 continued*

shown as solid lines, and the corresponding signals for the oligomeric protein(s) are shown as dashed lines. From both sets of data, the kinetics of FRET efficiency change for all five FRET pairs (**F–J**) was calculated (filled and empty green circles). The black lines through the data are a guide to the eye. The error bars in the fluorescence measurements are standard deviation of the mean, determined from four to five independent measurements, on separate samples. The error bars in the FRET efficiency were determined by propagating the errors in the fluorescence measurements.

DOI: https://doi.org/10.7554/eLife.44698.010

The following source data and figure supplements are available for figure 4:

**Source data 1.** Raw data for *Figure 4A–J*.
DOI: https://doi.org/10.7554/eLife.44698.016
**Figure supplement 1.** Concentration dependence of co-oligomerization monitored by fluorescence intensity.
DOI: https://doi.org/10.7554/eLife.44698.011
**Figure supplement 2.** Effect of doping ratio on fluorescence-monitored kinetics of co-oligomerization.
DOI: https://doi.org/10.7554/eLife.44698.012
**Figure supplement 3.** FRET monitored site-specific misfolding kinetics with W197-C223-DNP, W197-C169-DANS and W197-C223-DANS.
DOI: https://doi.org/10.7554/eLife.44698.013
**Figure supplement 4.** Time-resolved fluorescence anisotropy measurements to estimate $\kappa^2$.
DOI: https://doi.org/10.7554/eLife.44698.014
**Figure supplement 5.** Correlation analysis of global misfolding rates with thermodynamic stability and intrinsic physico-chemical properties, and measurement of local stability by FRET ratio and co-oligomerization kinetics by steady-state anisotropy.
DOI: https://doi.org/10.7554/eLife.44698.015

(*Figure 4—figure supplement 1*). In contrast, when a doping ratio of 1:99 was used (while keeping the total protein concentration fixed), the characteristic times remained unchanged for both the unlabelled W144-C153 and W197-C223 mutant variants (*Figure 4—figure supplement 2*). These results indicated that the slow change in Trp fluorescence also accompanied the formation of co-oligomers, irrespective of the position of Trp in the monomer.

In contrast, the burst phase change in Trp fluorescence, which was seen when a W197-containing mutant variant was the dopant protein, was complete before co-oligomer formation had commenced, with an amplitude which was independent of the total protein concentration, suggesting that it was an intramolecular change (*Figure 4—figure supplement 1*).

**Table 1.** Summary of the characteristic times of misfolding/oligomerization monitored by far-UV CD, tryptophan fluorescence and site-specific intra-molecular FRET

| Dopant protein | Characteristic time of fluorescence change (h) | Characteristic time of FRET change (h) |
|---|---|---|
| W144-C153 | 8.3 ± 1.3 | 7.1 ± 0.1 |
| W144-C153-TNB | 10.0 ± 1.0 | |
| W144-C199 | 6.7 ± 2.7 | 3.3 ± 0.7 |
| W144-C199-TNB | 6.3 ± 0.8 | |
| W144-C223 | 10.0 ± 1.0 | Not determinable |
| W144-C223-TNB | 5.9 ± 1.7 | |
| W197-C169 | 8.3 ± 0.7 | 1.7 ± 0.2, 16.7 ± 2.8 |
| W197-C169-TNB | 2.3 ± 0.1 | |
| W197-C223 | 8.3 ± 2.1 | 1.4 ± 0.3, 11.1 ± 2.5 |
| W197-C223-TNB | 2.9 ± 0.1 | |
| CD monitored misfolding* | 11.1 ± 3.7 | - |

[*]The characteristic time of the CD monitored misfolding was obtained by global fitting all the data in *Figure 3A*. Characteristic times were determined for a mixture of 2 mol% dopant protein and 98 mol% Trp-less moPrP. Error bars are standard deviation of the mean, determined from three to five independent measurements, on separate samples.
DOI: https://doi.org/10.7554/eLife.44698.017

## Site-specific conformational changes in moPrP monitored by intra-molecular FRET

Finally, the kinetics of intra-molecular changes across the structured CTD of moPrP were measured using FRET. Each unlabelled protein and its corresponding TNB-labelled counterpart was used as a dopant and was co-oligomerized separately with Trp-less moPrP at a 1:50 doping ratio. The amplitude and characteristic times of fluorescence change were both distinct for the corresponding labelled proteins. The data suggested that two of the distances monitored in these experiments, changed in a manner comparable to each other, but distinct from the other three distances (*Figure 4*).

Since changes in both distance R and Forster radius $R_0$ can affect the FRET efficiency, E, (*Equation 3*) it was important to consider both, while interpreting changes in E. It is to be noted that an increase in the quantum yield as well as a blue shift of the spectrum of the donor W144 is expected to increase $R_0$ (*Lakowicz, 2006*), although not to a significant extent, due to its 1/6th power dependence on both the quantum yield and the overlap integral (*Equation 4*). An increase in $R_0$ was expected to result in an increase in E, but for both FRET pairs W144-C153 and W144-C199, a decrease in E was observed. Consequently, the decrease in E reflected a true increase in distance. In contrast, for the TNB-labelled W197-C169 and W197-C223 mutant variants (*Figure 4*, Appendix 1, *Figure 4—figure supplement 3* and *Supplementary file 1*), the very small changes in quantum yield and overlap integral upon oligomerization result in only a 2.5–4% decrease in $R_0$ (*Equations 3 and 4*). The modest change in $R_0$ could only account for 8–23% of the total change in FRET efficiency observed for these proteins, suggesting that the decrease in E must be a consequence of a true change in distance. Moreover, a change in $\kappa^2$ also could not account for the observed change in FRET efficiency (Appendix 1, *Figure 4—figure supplement 4* and *Supplementary file 5*).

For the W144-C153 FRET pair, designed to monitor conformational changes in α1, the intra-molecular FRET efficiency decreased by ~22% with a characteristic time of 7.1 ± 0.1 hr (*Figure 4F* and *Table 1*). Among the two FRET pairs designed to monitor subdomain separation, W144-C199 decreased by ~54% with a characteristic time of 3.3 ± 0.7 hr (*Figure 4G* and *Table 1*), whereas W144-C223 did not show an observable change in intra-molecular FRET efficiency (*Figure 4H*). This could be because either the distance remained unchanged, or became greater than the sensitivity range (~11 Å to ~35 Å) of the Trp-TNB FRET pair.

The intra-molecular FRET efficiency for both FRET pairs with W197 as the donor fluorophore, W197-C169 and W197-C223 changed in a manner distinct from that of the other FRET pairs. First, the amplitude of the burst phase change in Trp fluorescence was comparable for both the unlabelled and labelled proteins, indicating that no change in intra-molecular FRET efficiency occurred in the dead time of measurement. This further confirmed that the burst-phase change in fluorescence reflected only a localized perturbation to the environment of W197, before oligomerization commenced.

Second, the FRET efficiency changed in two observable kinetic phases: a fast phase during which intra-molecular FRET efficiency increased and a slow phase during which it decreased. The apparent fast compaction of the sequence segments spanning both α2 and α3, as seen in the increase in FRET efficiency, appeared to take place with characteristic times of 1.7 ± 0.2 and 1.4 ± 0.3 hr, respectively (*Table 1*). The apparent fast increase in FRET efficiency, was more than 30-fold faster than the estimated timescale for the independent oligomerization of 2 μM labelled dopant protein (*Sabareesan and Udgaonkar, 2016*), and thus, the fast phase of FRET efficiency increase did not originate from (i) the oligomerization of 2 μM labelled dopant proteins independently of 98 μM Trp-less moPrP, or (ii) co-oligomerization of the two proteins on a faster timescale (see above). Furthermore, the fast compaction of the erstwhile helical sequence segments was also observed for a Trp-197-DNP-C2 FRET pair (with a $R_0$ of ~30 Å) and for a Trp-DANS FRET pair (Appendix 1 and *Figure 4—figure supplement 3*). The characteristic times for the slow phase of elongation/FRET decrease (16.7 ± 2.8 and 11.1 ± 2.5 hr) were comparable to the characteristic time of global misfolding, as monitored by far-UV CD (11.1 ± 3.7 hr) (*Figure 3A* and *Table 1*).

With the exception of W144-C153-TNB, all other TNB-labelled proteins, at 100 μM concentration, by themselves misfolded with characteristic times of ~1.5 hr, when monitored by CD (*Supplementary file 2*). Nevertheless, the oligomerization reactions in which these proteins were used (at 2 μM concentration) as dopants of 98 μM Trp-less moPrP took place with characteristic

times ranging between ~2 and 10 hrs, when monitored by fluorescence (*Figure 4* and *Table 1*). Moreover, even though the 100 µM DANS-labelled proteins by themselves misfolded with a characteristic time of ~0.5 hr (*Supplementary file 2*), which was ~2.5 fold faster than the corresponding TNB-labelled proteins, the observed characteristic times of fluorescence and FRET-monitored changes, when the DANS-labelled proteins were used as dopant proteins, were in qualitative agreement with those of the corresponding TNB-labelled proteins (*Figure 1—figure supplement 2*, *Figure 4—figure supplement 3* and *Supplementary file 2*). The lack of correlation between the timescales of misfolding of the labelled proteins by themselves, and timescales when the same proteins were used as dopant proteins, suggests that the fast changes in FRET efficiency are unlikely to be a consequence of altered co-oligomerization kinetics of 98 µM Trp-less moPrP when it co-oligomerizes with 2 µM dopant.

## Kinetic modelling of FRET and SEC data

Kinetic simulations and global fitting were carried out to test whether a parallel, sequential or triangular model best described the formation of $O_L$ and $O_S$ from M, taking into account both the SEC and FRET data. The criteria for choosing a suitable kinetic model were (i) it had to correctly predict the monomer loss kinetics determined from SEC; (ii) it had to correctly predict the concentrations of M, $O_L$ and $O_S$ at the end of 60 hr; and (iii) it had to correctly predict the experimentally observed FRET-monitored kinetics, by taking the sum of the FRET efficiencies of M, $O_L$ and $O_S$ weighted by their populations.

The parallel $O_S \leftrightarrow M \leftrightarrow O_L$ model and the sequential $M \leftrightarrow O_S \leftrightarrow O_L$ model could not satisfy all three criteria adequately, but the sequential $M \leftrightarrow O_L \leftrightarrow O_S$ model could do so. This model predicted that M oligomerizes reversibly to form $O_L$, which then disaggregates reversibly to form $O_S$, with characteristic times (1/k) for the $M \rightarrow O_L$, $O_L \rightarrow M$, $O_L \rightarrow O_S$ and $O_S \rightarrow O_L$ transitions, of 11.1, 20, 11.1 and 33.3 hr, respectively. It should be noted that in a previous study (*Jain and Udgaonkar, 2011*), $O_L$ had been shown to be capable of disaggregating to $O_S$. Global fitting of all the FRET data further predicted that (i) for the sequence segments 197–169 and 197–223, the FRET efficiency values are higher in $O_L$ than in $O_S$, and that (ii) for the 144–153 and 144–199 segments, the FRET efficiency values are lower in $O_L$ and $O_S$ than in M (*Figure 5* and *Table 2*). It should also be noted that in the kinetic models, it was assumed that all the transitions were first-order transitions. This assumption was made because the observed kinetics measured by CD, steady-state Trp fluorescence anisotropy, and by SEC, for all the unlabelled and labelled protein variants, were found to be describable well by a single exponential equation, even though only the two dissociation transitions ($O_L \rightarrow M$ and $O_L \rightarrow O_S$) are first-order transitions, while the association transitions are obviously not.

A triangular mechanism, in which both $O_S$ and $O_L$ can form reversibly from M, and $O_S$ and $O_L$ also equilibrate with each other, was also tested. Kinetic simulations according to the triangular mechanism also described all aspects of the data well, and all criteria were met. The forward and backward apparent rate constants for the $M \leftrightarrow O_S$ step, were however more than 10-fold slower than the all other apparent rate constants, indicating that negligible fraction of the $O_S$ oligomers formed directly from M. It is more appropriate to use the $M \leftrightarrow O_L \leftrightarrow O_S$ mechanism, because it is a simpler kinetic model than the triangular model.

## Discussion

The global stability, misfolding and oligomerization of WT moPrP is pH dependent. At pH 4 and 7, $\Delta G$ has been reported to be 4.6 and 6.04 kcal mol$^{-1}$ respectively. Moreover, it has been shown that while misfolding/oligomerization is 100% complete within 24 hr at pH 4, it is only ~5% complete at pH 5.7 on the same timescale (*Singh et al., 2014*; *Singh and Udgaonkar, 2016a*). Linear extrapolation to pH 7 suggests that misfolding/oligomerization should take years to complete at neutral pH. It should be noted that at pH 7, moPrP forms amyloid fibrils, and that oligomers are either completely absent or present in undetectable amounts (*Singh and Udgaonkar, 2013*).

Previous real-time NMR measurements had not detected any major structural rearrangement in the monomer, prior to oligomerization at pH 4 in the presence of 150 mM NaCl. This was supported by CD, SEC and HX-MS measurements of changes in conformation and size/population during oligomerization, under similar conditions, which had also indicated that major conformational change accompanying misfolding occurs only after, or concomitant with oligomerization. Among the two

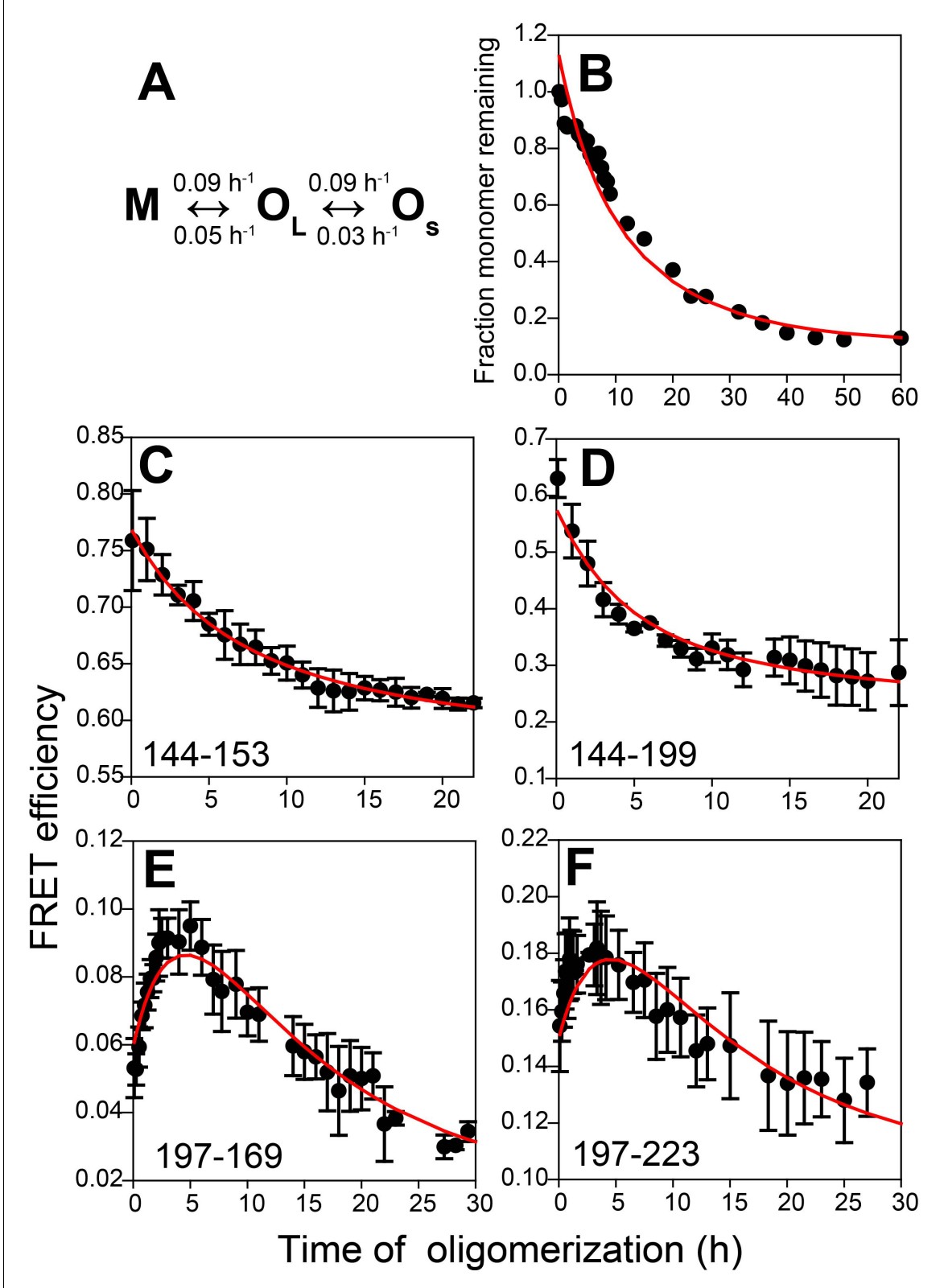

**Figure 5.** Kinetic model for misfolded oligomer formation of moPrP at pH 4 in the presence of 150 mM NaCl. (A) $O_L$ forms reversibly from M with forward and backward rate constants of $0.09 \pm 0.02$ and $0.05 \pm 0.01$ h$^{-1}$, respectively, and subsequently reversibly dissassembles to form $O_S$ with forward and backward rate constants of $0.09 \pm 0.03$ and $0.03 \pm 0.01$ h$^{-1}$, respectively. (B) Normalized monomer loss kinetics accompanying oligomerization of 100 μM Trp-less moPrP. The red line through the data is the fit of the experimental data to the model in (A). (C-F) Site-specific

*Figure 5 continued on next page*

*Figure 5 continued*

misfolding kinetics monitored by FRET efficiency as described in *Figure 4*. The red line through the data is the fit of the experimental data to the model in (**A**). The FRET efficiencies of M, $O_L$ and $O_S$ for each sequence segment, as estimated from the global fitting routine are tabulated in *Table 2*.
DOI: https://doi.org/10.7554/eLife.44698.018

tryptophan residues, W144 and W197 that were used as donor fluorophores in this study, W197-containing mutant variants exhibited a burst-phase change in fluorescence, when they were used as dopant proteins in co-oligomerization experiments. This is likely a consequence of a local change in monomeric moPrP before the start of oligomerization, namely the disruption of the K193-E195 salt-bridge in the α2-α3 loop, in accordance with previous NMR measurements (*Sengupta et al., 2017*).

To probe the major conformational changes in the monomeric unit as it misfolds into β-sheet-rich oligomers in real time with segment-specific resolution, FRET measurements were employed. The presence of eight tryptophan residues in WT moPrP prevented its use in suppressing the intermolecular contributions to the FRET signal during oligomer formation. Trp-less moPrP was therefore used as the pseudo-WT moPrP analogue for the FRET measurements. The secondary structure and size (as estimated from CD and DLS measurements, respectively) (*Sengupta and Udgaonkar, 2017*) of the Trp-less moPrP oligomers were found to be comparable to that of WT moPrP (*Figure 1—figure supplement 2*). Moreover, Trp-less moPrP formed oligomers L and S ($O_L$ and $O_S$) to similar extents as did WT moPrP at pH 4 in 150 mM NaCl. However, Trp-less moPrP misfolded and oligomerized ~2.5 fold slower compared to that of WT moPrP (*Figure 1—figure supplement 2* and *Supplementary file 2*). With this caveat in mind, the FRET measurements have allowed the delineation of the major structural changes that take place in the monomer, as it converts into soluble β-sheet-rich oligomers at pH 4.

A qualitative comparison of the characteristic times of all structural changes monitored by FRET suggests that a compaction of the segments spanning the α2 and α3 helices is the fastest change (*Table 1*). The separation of the β1-α1-β2 sub-domain from the α2-α3 subdomain and the conformational change in α1 appear to be slower. The slow decrease in FRET efficiency in α1 is likely to be due to the unfolding of this helix (*Singh and Udgaonkar, 2015a*). Remarkably, these results suggest that domain separation occurs spontaneously under acidic conditions which mimic the endocytic environment in the cell, in marked contrast to previous results where oligomerization was induced by thermal denaturation (*Eghiaian et al., 2007*). Moreover, this study shows that it is possible to directly show that the β1-α1-β2 sub-domain separates from the α2-α3 subdomain during the formation of the misfolded oligomers, by using an appropriately placed FRET donor-acceptor pair. In previous studies, the importance of subdomain separation had been inferred only indirectly from the observation that disulfide-crosslinking of the subdomains can abolish oligomerization (*Hafner-Bratkovic et al., 2011*). The slowest change appears to be the elongation of sequence segments that had spanned α2 and α3; this is possibly due to their conversion into β-sheets and is also reported on by the global probe CD (*Figure 6*).

**Table 2.** Summary of the FRET efficiencies of M, $O_L$ and $O_S$ obtained from global fitting of FRET data.

| Sequence segment | FRET efficiency (M) | FRET efficiency ($O_L$) | FRET efficiency ($O_S$) |
|---|---|---|---|
| 144–153 | 0.77 ± 0.01 | 0.50 ± 0.03 | 0.59 ± 0.02 |
| 144–199 | 0.57 ± 0.01 | 1e-7±1e-13 | 0.28 ± 0.03 |
| 197–169 | 0.07 ± 0.01 | 0.16 ± 0.01 | 1e-20 (constrained) |
| 197–223 | 0.17 ± 0.01 | 0.21 ± 0.02 | 0.06 ± 0.02 |

[*]The FRET efficiencies were obtained by global fitting the data from *Figure 4*. The FRET efficiencies of M, $O_L$ and $O_S$ was allowed to vary locally in each case, except in the case of sequence segment 197–169, where the FRET efficiency of $O_S$ had to be constrained to a low value of 1e-20 for an acceptable fit to the data. Errors in the FRET efficiency are standard errors from the fitting routine.
DOI: https://doi.org/10.7554/eLife.44698.019

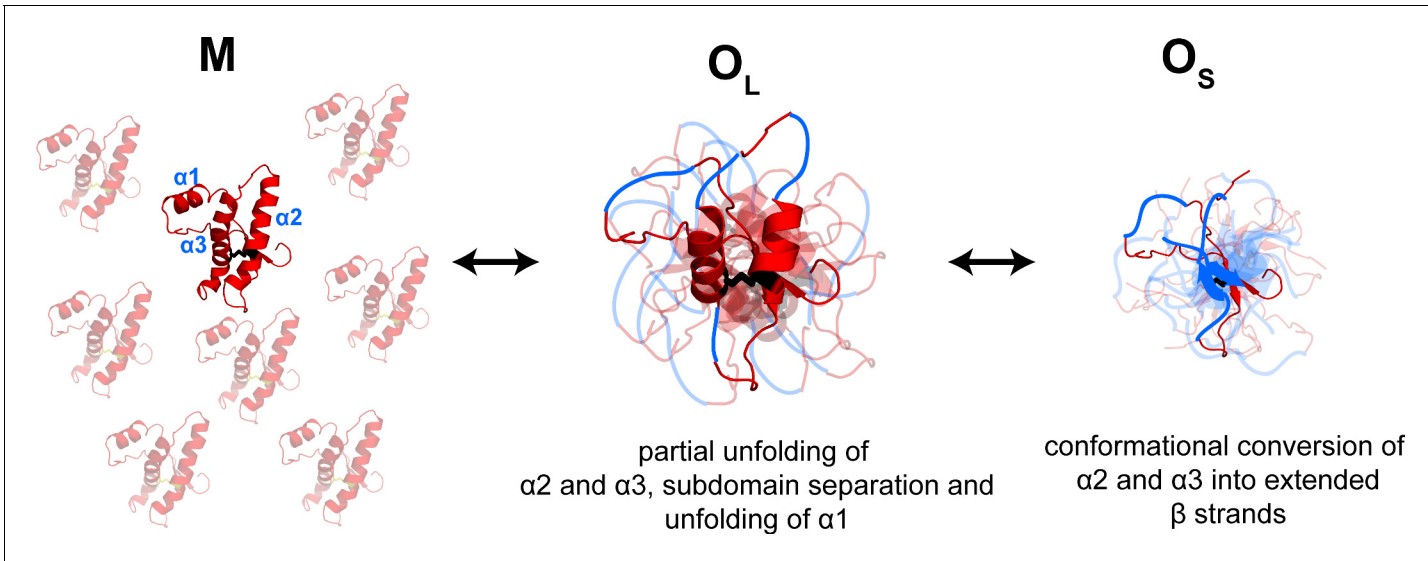

**Figure 6.** Model depicting the site-specific misfolding moPrP at during the course of oligomer formation at pH 4. The sequence segments spanning the α2 and α3 helices undergo a compaction as monomers convert reversibly to large oligomers, $O_L$. As oligomers $O_L$ disassemble reversibly to form small oligomers, $O_S$, the compact sequence segments spanning the erstwhile α2 and α3 helices elongate into extended β-strands. The α1-β1-β2 and α2-α3 subdomains are separated, and α1 is unfolded in both $O_L$ and $O_S$, but to variable extents. The NTR region is not shown for clarity. The disulphide bond is shown as black sticks. The transparent subunits in the growing oligomer represent the co-oligomerizing Trp-less moPrP which does not contribute to the FRET signal at any time during the oligomerization reaction. The figure is not drawn to scale.
DOI: https://doi.org/10.7554/eLife.44698.020

From the FRET measurements reported here, the sequence segment spanning α3 appears to undergo a fast compaction and a slow elongation, as does the sequence segment spanning α2. In contrast, HX-MS experiments could not detect any significant conformational change in this segment upon oligomer formation: α3 was found to be highly protected against HX in both the monomer and oligomer (*Sabareesan and Udgaonkar, 2016*; *Singh and Udgaonkar, 2015a*). HX-MS measurements probe the extent of exchange of backbone amide protons/deuterons with solvent and are silent to structural changes, which do not result in a measurable change in protection against HX. Since α3 is part of the buried core of both the monomer and the oligomer, HX-MS fails to detect conformational changes which might be taking place in α3, concomitant to oligomer formation. FRET, on the other hand, can detect these structural changes readily. This explains the apparent discrepancy between the HX-MS and FRET monitored conformational changes in α3 during the course of oligomer formation. Importantly, these results are in contrast to EPR studies carried out at neutral pH, on disulfide-free mutant variants of the PrP in nanodiscs, which had suggested that α1 and α3 retain their helicity, and that only α2 undergoes conversion to the β-conformation (*Yang et al., 2015*).

The observations from the SEC experiments that both large ($O_L$) and small ($O_S$) oligomers are formed sequentially from monomer M, and that both oligomers and monomer coexist at equilibrium at pH 4 in the presence of 150 mM NaCl, along with the previous demonstration that $O_L$ can disaggregate into $O_S$ (*Jain and Udgaonkar, 2010*) allowed kinetic modelling of the SEC data according to a M ↔ $O_L$ ↔ $O_S$ mechanism. The kinetic modelling and global fitting of the SEC and FRET data together yielded forward and backward rate constants for each step. Specifically and importantly, the biphasic (increase followed by decrease) changes in FRET efficiency observed for the sequence segments 197–169 and 197–223 were found to be a consequence of higher FRET efficiency values in $O_L$ than in M and $O_S$. In addition, the decrease in FRET efficiency observed for sequence segments 144–153 and 144–199 was found to be a consequence of lower FRET efficiency values in $O_L$ and $O_S$ than in M (*Table 2*).

Hence, the kinetic modelling indicates that the structural changes accompanying oligomerization occur in two steps. The formation of $O_L$ from M is accompanied by compaction of the sequence segments spanning the α2 and α3 helices, which accounts for the higher FRET efficiency seen for

sequence segments 197–169 and 197–223 in $O_L$. The subsequent dissociation of $O_L$ to $O_S$ is accompanied by an expansion of the same sequence segments, which is likely to be due to the formation of extended β-sheet structure in $O_S$. It is possibly the formation of β-sheet structure that makes $O_S$ more stable than $O_L$. The estimated FRET efficiency values for the sequence segments 144–153 and 144–199 further predict that α1 has unfolded and the erstwhile β1-α1-β2 sub-domain has separated from the erstwhile α2-α3 sub-domain to different extents in $O_L$ and $O_S$. These observations are supported by previous HX-MS measurements of the oligomers at pH 2 which showed that the sequence stretch 190–197 spanning a segment of the α2 helix and the α2- α3 loop is weakly protected in $O_L$, but moderately protected in $O_S$, suggesting that the α2 helix is unstructured in $O_L$, but may have converted to β-sheet in $O_S$ (*Singh et al., 2012*). Moreover, the α1 helix is weakly protected in $O_L$, but moderately protected in $O_S$ in accordance with the FRET results.

For an α-helix to convert into a β-sheet, intra-helical hydrogen bonds must be disrupted, for new inter-strand hydrogen bonds to form. It has been suggested that helices must undergo partial or complete unfolding before they can re-arrange their hydrogen bond structure to form β-sheets (*Ding et al., 2003*; *Qin and Buehler, 2010*). In the case of moPrP, hydrogen bonding at both the ends and/or middle of α2 and α3 can dissolve, but due to the presence of the native disulfide bond, residual structure will still be present. This increase in dynamics, without the complete loss of structure might allow the two ends of the helices to come closer, leading to the modest increase in FRET efficiency, distinct from the random coil structure of completely unfolded segments, which typically have lower FRET efficiencies than their folded counterparts. Subsequently, the formation of new hydrogen bonds between β-strands within and between monomers (leading to a decrease in FRET efficiency as β-strands are usually longer than α-helices) is complete within a timescale that is similar to that of global misfolding monitored by far-UV CD. The presence of highly dynamic and frustrated sequence segments like the TVTTTT stretch at the C terminal end of α2, possibly aids the early partial unfolding/compaction (*Chen and Thirumalai, 2013*). On the other hand, the disulphide bond stapling α2 and α3 might be a deterrent to complete unfolding, but could be crucial in positioning the two helices in an optimal position for inter-strand hydrogen bond formation. Alternatively, the compaction could be also be a result of non-native hydrogen bond formation which neither mimics the α-helix or β-sheet arrangement. Indeed, folding simulations of α-helix formation have detected misfolded β-hairpin like structures and compact structures with non-native hydrogen bonds (*Bertsch et al., 1998*; *Lin et al., 2014*).

In conclusion, the data presented here is the first real-time experimental demonstration of the sequence of segment-specific conformational changes that occur in each monomeric unit of moPrP as it forms oligomers. It will be important to establish whether the compaction-elongation mechanism of the α to β switch, delineated here for moPrP at pH 4, is shared by other proteins that undergo a similar conformational change during aggregation. Finally, the search for new molecules with the potential to completely abolish misfolding can benefit greatly from studies probing the effect of pathogenic mutations and anti-prion drugs on each of the multiple conformational changes delineated in this study, which lead to global misfolding.

# Materials and methods

## Key resources table

| Reagent type (species) or resource | Designation | Source or reference | Identifiers | Additional information |
|---|---|---|---|---|
| Recombinant DNA reagent | Trp-less-moPrP (pET22b vector) | https://doi.org/10.1016/j.pep.2017.07.014 | NA | Novagen-Sigma Aldrich |
| Recombinant DNA reagent | W144-C153-moPrP (pET 22b vector) | https://doi.org/10.1016/j.pep.2017.07.014 | NA | Novagen-Sigma Aldrich |
| Recombinant DNA reagent | W144-C199-moPrP (pET22b vector) | https://doi.org/10.1016/j.pep.2017.07.014 | NA | Novagen-Sigma Aldrich |
| Recombinant DNA reagent | W144-C223-moPrP (pET22b vector) | https://doi.org/10.1016/j.pep.2017.07.014 | NA | Novagen-Sigma Aldrich |

*Continued on next page*

*Continued*

| Reagent type (species) or resource | Designation | Source or reference | Identifiers | Additional information |
|---|---|---|---|---|
| Recombinant DNA reagent | W197-C169-moPrP (pET22b vector) | https://doi.org/ 10.1016/j.pep.2017.07.014 | NA | Novagen-Sigma Aldrich |
| Recombinant DNA reagent | W197-C223-moPrP (pET22b vector) | https://doi.org/ 10.1016/j.pep.2017.07.014 | NA | Novagen-Sigma Aldrich |
| Peptide, recombinant protein | Trp-less-moPrP | https://doi.org/ 10.1016/j.pep.2017.07.014 | NA | |
| Peptide, recombinant protein | W144-C153-moPrP | https://doi.org/ 10.1016/j.pep.2017.07.014 | NA | |
| Peptide, recombinant protein | W144-C199-moPrP | https://doi.org/ 10.1016/j.pep.2017.07.014 | NA | |
| Peptide, recombinant protein | W144-C223-moPrP | https://doi.org/ 10.1016/j.pep.2017.07.014 | NA | |
| Peptide, recombinant protein | W197-C169-moPrP | https://doi.org/ 10.1016/j.pep.2017.07.014 | NA | |
| Peptide, recombinant protein | W197-C223-moPrP | https://doi.org/ 10.1016/j.pep.2017.07.014 | NA | |
| Software, algorithm | DynaFit | https://doi.org/ 10.1006/abio.1996.0238 | NA | |

## Reagents

All reagents used for experiments were of the highest purity grade from Sigma, unless otherwise specified. Urea was purchased from USB, and GdnHCl for protein purification from HiMedia. 2,4 DNP (dinitrophenyl C2 maleimide) was purchased from Anaspec.

## Protein expression and purification

Five single Trp, single Cys-containing constructs, and a Trp-less construct were used in these experiments. The cloning, expression and purification of these proteins have been described elsewhere (*Sengupta and Udgaonkar, 2017*). The correct mass of all the constructs was verified by ESI-MS mass spectrometry (Appendix 1, *Figure 1—figure supplement 1*).

## Labelling of single Trp, single Cys-containing moPrP with non-fluorescent acceptor thionitro benzoate (TNB), dinitrophenyl (DNP) or fluorescent 5-((((2-Iodoacetyl)amino)ethyl)amino)Naphthalene-1-Sulfonic Acid) (DANS)

Briefly, purified single Trp, single Cys-containing mutant variants were reduced with a 10-fold excess of TCEP, for 12 hr at 4℃ under native conditions (in 10 mM NaOAc, pH 4) to remove any glutathione covalently linked to the extra cysteine. For a buried cysteine such as C153, the reduction reaction on the native protein was carried out for 36 hr. The protein was then either purified a second time by cation-exchange chromatography and dialysis against MQ water at 4℃ to yield the unlabelled mutant variant, or processed further for labelling with TNB or DNP.

The protein, after reduction, was diluted with 20 mM Tris, pH 7.5, such that the final concentration of the protein during labelling was no more than 0.5 mg/ml. This was followed by the drop-wise addition of 50-fold excess of DTNB in 20 mM Tris, pH 7.5, while continuously stirring. For labelling with 2,4-DNP, a concentrated stock solution of 2,4-DNP in DMSO was added drop-wise to the dilute protein solution, in 20 mM Tris, pH 7.5, while stirring. The labelling reaction was allowed to proceed for 12 to 36 hr at 4℃ (depending upon the extent of burial of the cysteine residue). For labelling with IAEDANS, a total protein concentration of 25 μM and a 1.5–2 fold excess of dye was used, to avoid non-specific labelling of lysines. Some protein was found to precipitate during the labelling reaction, which was removed by centrifugation at 18,000 rpm. The supernatant, containing the labelled protein of interest was purified with a 5 ml FF-CM Sepharose (GE Healthcare) cation-exchange column. This was followed by extensive dialysis against MQ water after which the protein was flash-frozen and stored at −80℃ until further use. The extent of labelling with TNB/DNP/DANS

was found to be ≥95% for all but one mutant variant, from ESI-MS mass spectrometry (Appendix 1, *Figure 1—figure supplement 1*). The only exception was W197-C223-DANS, which was labelled to an extent of 85%. All labelled proteins were found to have an intact native disulphide bond. Only one labelled moiety was found to be present on each of the labelled mutant variants. The concentration of the labelled proteins was estimated from their absorbance at 280 nm, after correcting for the contribution of the dye (estimated from the absorbance) as described in the manufacturer's protocol.

## Equilibrium unfolding monitored by far-UV CD

Urea-induced equilibrium unfolding of all mutant variants in their unlabelled and labelled forms at pH 4, 25°C was monitored by far-UV CD at 222 nm. The corresponding thermodynamic stability, ΔG (kcal mol$^{-1}$) and mid-point of unfolding, $C_m$ (M) was obtained by fitting the fraction unfolded data versus denaturant concentration to a two-parameter equation (*Agashe and Udgaonkar, 1995*).

## DLS measurements of WT and Trp-less oligomers

WT and Trp-less moPrP were oligomerized to form misfolded β-sheet-rich oligomers at pH 4 in the presence of 150 mM NaCl. The oligomers were diluted with 1X aggregation buffer to a final concentration of 10 μM for DLS measurements (*Figure 1—figure supplement 2*). DLS measurements were carried out as described earlier (*Sengupta and Udgaonkar, 2017*).

## Global misfolding kinetics monitored by far-UV CD

The global misfolding kinetics of all unlabelled, labelled and Trp-less moPrP mutant variants were monitored by recording the far-UV CD signal at 228 nm. Misfolding and oligomerization were initiated by the addition of 10x aggregation buffer to monomeric protein initially present in 10 mM sodium acetate at pH 4. The final protein concentration was 100 μM, and the final buffer composition was 10 mM sodium acetate, 150 mM NaCl, pH 4 (1x aggregation buffer), 37°C. At each kinetic time point, the mixture of oligomers and monomers was diluted with 1x aggregation buffer, such that the protein concentration during measurement was 10 μM (*Sengupta et al., 2017*). The data were fit to a single exponential equation

$$y = y_0 + a\left(1 - e^{-bt}\right) \tag{1}$$

to obtain the apparent rate-constants (*Supplementary file 2*).

## Co-oligomerization with Trp- less moPrP to suppress inter-molecular contributions to FRET

Since intra-molecular conformational changes were to be monitored by FRET, it was necessary that all inter-molecular FRET be effectively suppressed. To achieve this, co-oligomerization of unlabelled and TNB-labelled W144-C153 moPrP variant with Trp-less moPrP in different doping ratios was employed. The total protein concentration in each oligomerization reaction was kept fixed at 100 μM. The concentration of the dopant, W144-C153 moPrP, was systematically varied from 100 to 2 mol%.

## Misfolding kinetics of Trp-less moPrP doped with varying concentrations of W197-C223-TNB

To verify that at the low labelling densities employed in these experiments, the dopant and Trp-less protein form misfolded co-oligomers at a rate which is indistinguishable from the global misfolding kinetics of Trp-less moPrP alone, misfolding of Trp-less moPrP with dopant concentrations of 1, 2, 10 and 20 mol% (while keeping total protein concentration fixed at 100 μM) were measured by monitoring their misfolding kinetics by measuring the far-UV CD signal at 228 nm. The dopant, W197-C223-TNB was chosen because it misfolds almost ~10 fold faster by itself as compared to Trp-less moPrP.

## Misfolding kinetics of Trp-less moPrP doped with 2 mol% of dopant

All unlabelled and TNB-labelled single Trp, single Cys-containing mutant variants (dopant proteins) were co-oligomerized with Trp-less moPrP, such that the total protein concentration was 100 μM

and the dopant protein concentration in each reaction was 2 µM. The global misfolding kinetics was monitored by far-UV CD as described earlier.

## Local stability measured using FRET ratio for the DANS-labelled mutant variants

Urea-induced local unfolding of W197-C169-DANS and W197-C223-DANS at pH 4˚C and 25˚C was monitored by monitoring the ratio of fluorescence intensities at 495 nm and 345 nm ($F_{495}$/ $F_{345}$), respectively, which are the intensity maxima for the IAEDANS and tryptophan fluorophores respectively, when the sample is excited at 295 nm (*Figure 4—figure supplement 5*). The excitation and emission slit widths were set at 1 and 5 nm respectively. The same samples were also used for the CD measurements (*Figure 1—figure supplement 2*). The corresponding local stability, ΔG (kcal mol$^{-1}$) and mid-point of unfolding, $C_m$ (M) was obtained by fitting the normalized data versus denaturant concentration to a six-parameter equation (*Agashe and Udgaonkar, 1995*).

## Co-oligomerization kinetics of Trp-less moPrP doped with 2 mol% of dopant monitored by steady-state fluorescence anisotropy

W144-C153-TNB and W197-C223-TNB (dopant proteins) were co-oligomerized with Trp-less moPrP, such that the total protein concentration was 100 µM and the dopant protein concentration in each reaction was 2 µM. The co-oligomerization kinetics was monitored by steady-state tryptophan fluorescence anisotropy at an excitation wavelength of 295 nm and emission wavelength of 340 nm on a Fluorolog spectrofluorimeter with an excitation slit width and emission slit width of 5 nm respectively, and corrected using the G-factor determined for the instrument, every day (*Figure 2C*). The data were fit to a single exponential equation (*Equation 1*) to obtain the rate-constants (*Supplementary file 4*).

## Oligomerization kinetics monitored by size-exclusion chromatography

100 µM Trp-less moPrP was oligomerized as described above. At each time point, 25 µL of the oligomerization mixture was mixed with 225 µL of 1X aggregation buffer, such that the total protein concentration was 10 µM. 200 µL of this mixture was injected into a Waters Protein Pak 300-SW column and the oligomerization kinetics monitored and the data analysed as described earlier (*Sabareesan and Udgaonkar, 2016*).

## Kinetics of misfolding monitored by FRET

Every pair of unlabelled and labelled single Trp, single Cys-containing mutant variants was either individually co-oligomerized with Trp-less moPrP, or in its absence. The total protein concentration was fixed at 100 uM, with the dopant concentration at 2 µM for the former case, and only 2 µM for the latter case, in a reaction volume of 500 µL. All measurements were made on a Fluoromax 4 spectrofluorimeter. A quartz cuvette of path length 2 × 10 mm was used for all measurements. The temperature of the reaction was maintained at 37˚C, with a circulating water bath. The cuvette was kept stoppered during the reaction to prevent solvent loss due to evaporation.

To eliminate the contribution of scatter to the fluorescence signal, a 325 nm long-pass filter was kept between the thermo-statted cuvette and the emission monochromator. An excitation wavelength of 295 nm, an emission wavelength of 340 nm, excitation slit width of 1 nm and emission slit width of 5 nm were used in all experiments. The kinetics mode of the Fluoresscence software was used for acquisition for most experiments. The number of cycles and time interval between cycles were set according to the number of kinetic data points and total acquisition time of the reaction. Every kinetic time point was an average of 10 points acquired over 10 s. The anti-photobleaching mode was activated in these experiments, such that the shutter remained closed between acquisitions, so that negligible loss of fluorescence intensity due to photobleaching occurs. Every experiment was repeated 4–5 times on an average. For some experiments, kinetic time points were acquired manually, at desired intervals. A control experiment was carried out using the same acquisition parameters, but in the presence of native buffer only (10 mM sodium acetate, pH 4). The signal was found to remain constant over a period of 30 hr indicating no photo-bleaching was taking place (Appendix 1, *Figure 4—figure supplement 1A*).

The fluorescence emission spectra of the protein(s) were recorded before and every oligomerization reaction. First, the two proteins were mixed such that the concentration of Trp- less moPrP and dopant were 109 and 2.2 µM, respectively, in 10 mM sodium acetate, pH 4 buffer. This was incubated for 5 min in the thermostatted cuvette before recording the spectrum. The fluorescence intensity of the monomer was adjusted for concentration by multiplying the measured value at 340 nm by 0.9. These concentrations were chosen such that after addition of 50 µL of 10x aggregation buffer to 450 µL of the above protein mixture, the final concentrations of Trp-less moPrP and dopant would be 98 µM and 2 µM, respectively.

Similarly, before recording the first kinetic time point after initiation of misfolding, a dead time of 5 min was allowed for equilibration to 37°C. For background correction, the intensity corresponding to 98 µM Trp-less moPrP in the same buffer conditions was subtracted from each data point in the doped samples. The background value was found not to change with time, showing that contribution due to scatter was indeed negligible, and that the Trp-less protein preparation was free from tryptophan contamination. The data were normalized to the signal for the corresponding donor-only sample at t = 0, $F_D(0)$, for all samples. FRET at every kinetic time point was calculated according to the following formula:

$$E(t) = 1 - \frac{F_{DA}(t)}{F_D(t)} \tag{2}$$

where $F_D$ and $F_{DA}$ are the fluorescence emission intensity values for the donor-only, and donor acceptor sample, respectively, at each kinetic time point t.

FRET efficiency is described by the equation:

$$E = \frac{1}{1 + (R/R_0)^6} \tag{3}$$

where R is the distance between the donor and the acceptor and the Foster Radius, $R_0$ is the distance, at which E is 0.5.

Foster Radius, $R_0$ is described by the equation:

$$R_0 = 0.211 \cdot \left( \kappa^2 \cdot QY \cdot n^{-4} \cdot J(\lambda) \right)^{\frac{1}{6}} \tag{4}$$

where $\kappa^2$ is the orientation factor between the donor and acceptor dipoles, QY is the quantum yield of the donor, n is the refractive index of the medium and $J(\lambda)$ is the overlap integral between the fluorescence emission spectrum of the donor and the absorption spectrum of the acceptor (*Lakowicz, 2006*). Similar fluorescence measurements were made for the 1:99 samples corresponding to doping with 1 mol% dopant protein.

## Global fitting to parallel and sequential models of oligomerization

Global fitting was carried out with the program DynaFIT (*Kuzmic, 1996*). Briefly, parallel and sequential reaction schemes were tested for the reversible formation of oligomers $O_s$ and $O_L$ from M. The ratios of forward and backward rate constants were fixed to account for the ratio of $O_s/O_L \sim 3$ and the amounts of M, $O_L$ and $O_s$ that was approximately estimated from the SEC data. The FRET and SEC data were globally fit separately to first test which of the reaction schemes best describes both data sets with comparable rate constants. While the M ↔ $O_S$ ↔ $O_L$ and the $O_S$ ↔ M ↔ $O_L$ mechanisms were able to fit the FRET and SEC data separately to satisfaction, global fitting of both data sets together did not yield acceptable fits. All parameters were allowed to vary in the fitting procedure except for the FRET efficiency value of $O_S$ for the 197–169 sequence segment, which was constrained to a low value of 1e-20 while global fitting of FRET data to the M ↔ $O_L$ ↔ $O_S$ mechanism. While fitting the normalized monomer loss kinetics from the SEC data, the signal for M was allowed to vary, while the signal for $O_S$ and $O_L$ were fixed to zero.

## Acknowledgements

We thank members of our laboratory for critically reading the manuscript, and MK Mathew, S Gosavi, A Singh and S Krishna for helpful discussions. We thank Sandhya Bhatia for help with the

time-resolved anisotropy measurements and analysis. We thank Harish Kumar for collecting the DLS data for the moPrP oligomers. We thank Roumita Moulick for providing some of the constructs used in this study. We thank Prashant N Jethva, Pooja Malhotra and Rupam Bhattacharya for the ESI-MS measurements. JBU is a recipient of a JC Bose National Fellowship from the Government of India, and IS is a recipient of the Innovative Young Biotechnologist Award from the Department of Biotechnology, Government of India. This work was funded by the Tata Institute of Fundamental Research and by the Department of Biotechnology, Government of India.

## Additional information

### Funding

| Funder | Grant reference number | Author |
| --- | --- | --- |
| Department of Biotechnology, Ministry of Science and Technology | JC Bose Fellowship | Jayant Udgaonkar |
| Department of Biotechnology, Ministry of Science and Technology | IYBA | Ishita Sengupta |
| Tata Institute of Fundamental Research | | Jayant Udgaonkar |

The funders had no role in study design, data collection and interpretation, or the decision to submit the work for publication.

### Author contributions

Ishita Sengupta, Conceptualization, Resources, Data curation, Formal analysis, Funding acquisition, Validation, Investigation, Visualization, Methodology, Writing—original draft, Project administration, Writing—review and editing; Jayant Udgaonkar, Conceptualization, Resources, Software, Supervision, Funding acquisition, Writing—original draft, Project administration, Writing—review and editing

### Author ORCIDs

Ishita Sengupta (iD) https://orcid.org/0000-0002-2679-6954
Jayant Udgaonkar (iD) https://orcid.org/0000-0002-7005-224X

### Decision letter and Author response

Decision letter https://doi.org/10.7554/eLife.44698.029
Author response https://doi.org/10.7554/eLife.44698.030

## Additional files

### Supplementary files

• Supplementary file 1. Parameters used for the estimation of FRET efficiency in monomeric moPrP mutant variants.
DOI: https://doi.org/10.7554/eLife.44698.021

• Supplementary file 2. Thermodynamic parameters obtained from urea-induced equilibrium unfolding of different moPrP variants at pH 4 and their misfolding rate constants monitored by far-UV CD at 228 nm, for 100 µM protein.
DOI: https://doi.org/10.7554/eLife.44698.022

• Supplementary file 3. Misfolding rate constants for Trp-less moPrP alone, 197–223-TNB alone and Trp-less moPrP doped with 1 mol %, 2 mol %, 10 mol % and 20 mol % of 197–223-TNB, monitored by far-UV CD at 228 nm. The total protein concentration in each case was 100 µM.
DOI: https://doi.org/10.7554/eLife.44698.023

• Supplementary file 4. Apparent rate constant of co-oligomerization monitored by steady-state tryptophan fluorescence anisotropy at different dopant concentrations.
DOI: https://doi.org/10.7554/eLife.44698.024

• Supplementary file 5. Rotational correlation times, amplitudes and fundamental anisotropy r(0) values derived from fitting the time-resolved anisotropy data of 1,5-IAEDANS-labelled W197-containing mutant variants in the monomeric and oligomeric forms.
DOI: https://doi.org/10.7554/eLife.44698.025

• Transparent reporting form
DOI: https://doi.org/10.7554/eLife.44698.026

### Data availability

All data generated or analysed during this study are included in the manuscript and supporting files. Source data files have been provided for Figures 2, 3 and 4.

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

## Appendix 1

DOI: https://doi.org/10.7554/eLife.44698.027

# Supplementary materials

## Estimation of range of $\kappa^2$ values for the oligomeric forms of W197-containing mutant variants

Since it is not possible to measure the exact value of $\kappa^2$, a range of $\kappa^2$ values for both W197-containing mutant variants, W197-C169 and W197-C223 in their oligomeric forms were determined from time-resolved anisotropy measurements (*Figure 4—figure supplement 4* and *Supplementary file 5*). TNB is non-fluorescent; therefore, the proteins were labelled with 1,5-IAEDANS (see Materials and methods section) and measurements were made as described in *Jha and Udgaonkar (2009)*.

The estimated $\kappa^2$ values range from 0.4 to 2 and 0.21 to 2.5 for the proteins W197-C169 and W197-C223 respectively in their oligomeric forms. These values correspond to $R_0$ values ranging from 23.8 to 31.1 and 21.3 to 32.2 Å respectively (*Equation 4* in main text). If the FRET efficiency changes were only due to a change in $R_0$ without any change in distance R, the expected FRET efficiency values should range from 0.04 to 0.25 and 0.05 to 0.36 for the W197-C169 and W197-C223 proteins in their oligomeric forms respectively. The lower observed FRET efficiencies of 0.01 and 0.035 respectively for these proteins (*Figure 4—figure supplement 3*) cannot be a consequence of a change in $\kappa^2$ and $R_0$ solely, and must reflect a true change in distance R.

Moreover, the fundamental anisotropy values of 0.24 and ~0.28 respectively estimated from the fit (r(0)) are significantly lower than the fundamental anisotropies of both donor tryptophan ($r_0$ = 0.4) and acceptor 1,5-IAEDANS ($r_0$ = 0.36) groups determined in frozen solution, suggesting a significant amplitude (40–50%) for the free motion of both donor and acceptor probes (*Figure 4—figure supplement 4* and *Supplementary file 5*), even when attached to a large oligomeric species (*Lakowicz, 2006*; *Lakshmikanth et al., 2001*).

Therefore, (i) the fast sub-ns segmental motions of 40–50% amplitude of both donor and acceptor groups in the oligomeric forms of both proteins, (ii) the smaller size of the TNB label compared to the DANS label (iii) the inability of a change in $\kappa^2$ and $R_0$ to account for the observed FRET efficiency values in the oligomeric forms all justify the use of $\kappa^2$ = 2/3 for the oligomeric forms.

## Effect of mutation and labelling on secondary structure, stability and global misfolding kinetics

The secondary structure, thermodynamic stability and global misfolding kinetics of all the unlabelled and TNB/DANS-labelled mutant variants were determined experimentally using far-UV CD (*Figure 1—figure supplement 2* and *Supplementary file 2*). For the WT moPrP, the values reported here agree well with previously reported values at pH 4 of 4.1 kcal mol$^{-1}$, 1.2 kcal mol$^{-1}$ M$^{-1}$ and 3.5 M for $\Delta$G, m and $C_m$ respectively (*Cereghetti et al., 2003*). Most mutations in moPrP, which have been shown to alter the thermodynamic stability or folding/unfolding kinetics, have typically been located in buried positions in the α2-α3 subdomain, and involve hydrophobic side-chains (*Hart et al., 2009*). Keeping this in mind, all mutations were made in solvent-exposed positions, except for the completely buried M153C mutation in α1 and the partially-buried F197W mutation in the loop between α2 and α3 helices. The unlabelled W197-C169 and W197-C223 mutant variants used here were not destabilized, supporting earlier observations that the F197W mutation does not perturb stability and/or unfolding/folding kinetics despite it being equivalent to residue F198 in human PrP, a site for the pathogenic mutation F198S associated with the inherited prion disease, the GSS syndrome (*Jenkins et al., 2008*). However, the W197-C169 mutant variant misfolded with slightly faster kinetics, compared to the WT moPrP, possibly due to its location in the loop between β2 and α2 (*Agarwal et al., 2015*; *Gossert et al., 2005*). Somewhat counter intuitively, the completely

buried M153C mutation in α1 did not alter thermodynamic stability or global misfolding kinetics. The only mutant variant with a significant reduction in thermodynamic stability (but without a change in global misfolding kinetics) was W144-C199, (despite E199C being a solvent-exposed position) possibly due to an altered surface charge. A change in intrinsic properties of the amino acid sequence (for e.g. hydrophobicity, β-sheet propensity or charge) upon mutation could not account for the change in the global misfolding kinetics (*Figure 4—figure supplement 5*).

In marked contrast to the unlabelled mutant variants, a striking correlation between the global misfolding kinetics and position of the TNB/DANS acceptor moiety was seen for the corresponding labelled mutant variants (with no correlation to thermodynamic stability) at 100% labelling density. When the solvent-exposed positions C169, C199 or C223, in the α2-α3 subdomain were covalently modified with TNB/DANS, global misfolding kinetics was accelerated. In contrast, when the buried position C153 on α1 was covalently modified with TNB, global misfolding kinetics remained unchanged. The enhanced kinetics is unlikely to be due to a change in local stability of the α2 and α3 helices upon labelling (*Figure 4—figure supplement 5* and *Supplementary file 2*), but instead appears to be a result of enhanced association kinetics at high labelling densities. This is possibly because the formation of misfolded oligomers of moPrP at pH 4 is rate-limited by association (*Sengupta et al., 2017*), and involves inter-molecular interactions within the α2-α3 subdomain of monomeric PrP, which misfolds to form the β-rich core of the oligomers.

## Correlation of global misfolding rates with change in intrinsic properties, local and global thermodynamic stability upon mutation/labelling

An increase in protein misfolding/aggregation rates have been shown to correlate well with an increase in hydrophobicity, β-sheet propensity, population of aggregation-prone intermediates (*Chiti et al., 2003*; *Chiti et al., 2002b*; *Dobson, 2003*), a decrease in charge (*Calamai et al., 2003*; *Chiti et al., 2002a*) and/or local/global thermodynamic stability (*Chiti and Dobson, 2006*). The hydrophobicity and β-sheet propensities reported in *Chiti et al. (2003)* were used for this analysis. p-values >> 0.05 suggested that the global misfolding rate of the labelled and unlabelled mutant variants used in these studies have no significant correlation with any of these properties (*Figure 4—figure supplement 5*). Moreover, local stabilities of the α2 and α3 helices determined by the FRET ratio ($F_{495}/F_{345}$) in the corresponding DANS-labelled mutant variants (*Figure 4—figure supplement 5* and *Supplementary file 2*) was found to be in good agreement with the mean $\Delta G_{op}$ of ~3.4 ± 0.7 kcal mol$^{-1}$ and ≥2.6 ± 1.5 kcal mol$^{-1}$, across the slowest exchanging residues of the α2 and α3 helices, determined previously from HX-MS and NMR for WT moPrP (*Moulick et al., 2015*; *Singh and Udgaonkar, 2016b*). Therefore, a reduction of local stability was also not able to account for the faster misfolding rates for some mutant variants.

