## [Decision Letter]

Thank you for submitting your article "Monitoring site-specific conformational changes in real-time reveals the misfolding mechanism of the prion protein" for consideration by *eLife*. Your article has been reviewed by three peer reviewers, one of whom is a member of our Board of Reviewing Editors, and the evaluation has been overseen by John Kuriyan as the Senior Editor. The reviewers have opted to remain anonymous.

The reviewers have discussed the reviews with one another and the Reviewing Editor has drafted this decision to help you prepare a revised submission.

Summary:

In the present manuscript, Sengupta and Udgaonkar use FRET-based measurements to investigate conformational changes occurring in the α2-α3 and β-1-α1-β2 subdomains of the mouse prion protein (moPrP) after its incorporation into oligomers at low pH. Based on differential changes in FRET between Trp residues (W144 in α1 or W197 in the α2-α3 linker) and TNB attached to engineered Cys in 4 positions around those subdomains, the authors propose a model for moPrP misfolding in which the α2-α3 segment initially compacts, before separating from the β1-α1-β2 subdomain, followed by α1 unfolding and transition of the α2-α3 segment into extended non-native β sheets.

Revealing the mechanism by which PrP converts into a misfolded isoform is difficult and complex, yet very important, as PrP is the archetype for disease-related propagated misfolding. In contrast to previous studies that mostly analyzed the global aggregation process, the presented FRET-based experiments allow the investigation of sub-steps and conformational transitions in individual domains of PrP. Performing the measurements at low pH is physiologically relevant based on the hypothesis that misfolding in the endosome plays an important role in pathogenesis and/or propagation.

Essential revisions:

In their revised full submission, the authors responded in detail to initial criticism regarding potential changes in Foerster radius upon oligomerization and the artificially fast misfolding behavior of TNB-labeled variant. While the concerns about changes in Foerster radius, quantum yield or kappa square of TNB have been addressed to full satisfaction, there remain major questions about the misfolding kinetics of the TNB- or DANS-labeled mutants. All reviewers agreed that these significant technical concerns need to be addressed prior to further consideration of this manuscript for *eLife*.

1) There are major concerns regarding to what extent the mutations and labels are responsible for the observed FRET behavior, and whether misfolding of WT PrP indeed follows the same rules. Whatever leads to faster oligomerization of labeled mutant PrP in isolation may also influence its oligomerization with WT PrP and thus have an effect on the measured FRET kinetics. It is quite intriguing that the rate of initial FRET increase (0.6 or 0.7 h-1) agrees very well with the accelerated misfolding/oligomerization kinetics of the isolated labeled mutants (0.61 – 0.78 h-1). The presented controls (CD and anisotropy) are dominated by the oligomerization behavior of the excess unlabeled "WT" PrP and are thus inept to probe whether the labeled mutant PrP dopant oligomerizes with WT PrP at an accelerated rate, followed by slower assembly of additional WT PrP.

CD measurements at 10 or 20% dopant revealed a 50% higher global rate of misfolding when fit to a single exponential, which may in fact indicate faster initial association of TNB-labeled monomers even with WT PrP. Clear double exponential behavior may be hard to detect, given the limited resolution (Figure 2B) and the expected low amplitude of a fast phase when using excess WT PrP.

These concerns about mutant and/or label effects on the FRET measurements are reinforced by the presented model, proposing that the fast conformational change, i.e. helix compaction, occurs at a rate of 0.6-0.7 h-1 only *after* oligomerization, yet all non-FRET readouts monitoring this oligomerization (e.g. anisotropy) show rates of ~ 0.1 h-1. How is that possible? Maybe the helix compaction and segment separation actually happen *before* oligomerization, in a spontaneous misfolding event that precedes assembly? This question relates to the essential debate about whether natively folded PrP fluctuates into a non-native structure before joining an aggregate, or whether it joins the aggregate and then re-configures.

To draw more solid conclusions, the authors need to use a more direct technique (e.g. FCS, SEC, DLS, AFM and/or inter-molecular-FRET) for assessing the formation kinetics as well as size changes with time for the PrP variants, especially the Trp-less PrP. How do the amount and size of oligomers change with time? Measuring the diffusion constant (and hence size) by FCS as a function of time may allow the authors to assess whether the FRET-labelled molecules are changing conformation before or after becoming incorporated into the oligomer.

Alternatively, the authors could consider applying some size fractionation method at different time-points to see whether labelled PrP has joined an oligomer before or after (or during) the fast structural change. This approach would have the advantage that different fractions could be characterized regarding their FRET efficiency, providing extra information about how the structure changes along with the growth of the oligomer.

2) Another issue lies in the fact that mutating multiple Trp residues likely affects protein structure, stability, and dynamics (i.e. activation barriers), which raises questions about the robustness of the conclusions and their applicability. For instance, the Trp-less PrP shows the slowest kinetics, which may indicate that it has some deficiency in conformational conversion and oligomerization. As a minimum, the authors should use an additional method to verify whether the oligomer of Trp-less PrP has similar properties and formation efficiency as the WT oligomers. Whether the fluorescence/FRET measurement using TrP-less PrP in the dopant experiment really represent the kinetics and mechanism of the WT normal PrP is unclear. This issue is intrinsic to the approach and cannot be resolved for the current study, but should be discussed accordingly. An important aspect to consider for the W197 mutants, however, is that the equivalent residue in human PrP, F198, is the site of a pathogenic mutation. F198S is associated with GSS syndrome, an inherited prion disease. That the W197 mutants of moPrP show faster misfolding and oligomerization in the present study could therefore be related to the fact that misfolding is particularly sensitive to this residue, even though an F to W mutation is much more conservative than the F to S mutation responsible for GSS syndrome.

3) There were initial concerns among the reviewers about the number of novel findings and the apparent overall advance provided by this work. Several previous studies had already concluded that the α2-α3 and β1-α1-β2 segments have to separate and α1 has to unfold prior to the β-conversion, leaving the potential compaction of the α2-α3 helices as the only major new result. However, these concern could be somewhat alleviated during the reviewers' discussions and a more in depth review of the literature, which revealed that previous work, for instance by Eghiaian et al. and Hafner-Bratkovic et al. using thermal denaturation or disulfide crosslinking, were less conclusive and more questionable regarding their relevance for the 'natural' misfolding process of PrP. In contrast, the present paper shows that domain separation occurs spontaneously in acidic conditions, which are endogenously found in endosomes. The authors should therefore better emphasize these advances of their study and in more detail put them into perspective to previous conclusions. Moreover, the somewhat turgid and disjointed writing of the paper does not do full justice to the assays and results. The manuscript is also strongly dominated by the setup of the system and various controls, whereas critical experiments and their discussion are proportionally slim, which should be improved in a revised version.

---

## [Author Response]

Essential revisions:In their revised full submission, the authors responded in detail to initial criticism regarding potential changes in Foerster radius upon oligomerization and the artificially fast misfolding behavior of TNB-labeled variant. While the concerns about changes in Foerster radius, quantum yield or kappa square of TNB have been addressed to full satisfaction, there remain major questions about the misfolding kinetics of the TNB- or DANS-labeled mutants. All reviewers agreed that these significant technical concerns need to be addressed prior to further consideration of this manuscript for eLife.1) There are major concerns regarding to what extent the mutations and labels are responsible for the observed FRET behavior, and whether misfolding of WT PrP indeed follows the same rules. Whatever leads to faster oligomerization of labeled mutant PrP in isolation may also influence its oligomerization with WT PrP and thus have an effect on the measured FRET kinetics. It is quite intriguing that the rate of initial FRET increase (0.6 or 0.7 h-1) agrees very well with the accelerated misfolding/oligomerization kinetics of the isolated labeled mutants (0.61 – 0.78 h-1). The presented controls (CD and anisotropy) are dominated by the oligomerization behavior of the excess unlabeled "WT" PrP and are thus inept to probe whether the labeled mutant PrP dopant oligomerizes with WT PrP at an accelerated rate, followed by slower assembly of additional WT PrP.

The reviewers have raised an interesting point: the observation that the rate of initial FRET increase for the 197-169 and 197-223 segments in the case of (2% dopant + 98% Trp-less protein) is the same as the rate of the CD-monitored rate for 100% dopant, could, in principle, suggest that the 2% dopant could be driving the 98% Trp-less protein to behave like 98% dopant during co-oligomerization. We think this is highly unlikely because the average oligomer size is about 50, and would therefore have only one dopant molecule. Nevertheless, it is important to demonstrate directly that 1 dopant molecule per 50-mer during co-aggregation is not responsible for the “intriguing” observation.

The similarity of rates alluded to, is only true when the dopant molecule is TNB–labelled. We have also carried out FRET experiments for monitoring the 197-169 and 197-223 segments, where instead, the dopant molecules are DANS-labelled. 100% DANS-labeled dopant molecules misfolded at a CD-monitored rate of ~2 h^-1^, which was about 2.5 fold faster than 100% TNB-labelled dopant molecules. Yet, the characteristic times (1/k) for the fast and slow phases of FRET change were comparable for (2% TNB-labelled dopant + 98% Trp-less protein) and (2% DANS-labelled dopant + 98% Trp-Less protein. If the fast change in FRET was indeed a consequence of the altered oligomerization kinetics of Trp-less protein co-aggregating with 2 mol% 197-223-TNB or 197-169-TNB, then it should have been ~2.5 fold faster for the corresponding DANS-labelled dopants. This is not the case. Hence, the fast FRET change is not an effect of the altered oligomerization kinetics of 98 μM Trp-less protein in the presence of 2 μM 197-169-TNB/DANS or 197-223-TNB/DANS, but is a consequence of the compaction of these sequence segments.

We discuss this in the sixth paragraph of the subsection “Site-specific conformational changes in moPrP monitored by intra-molecular FRET” where we state: “Moreover, even though the 100 μM DANS-labelled proteins by themselves misfolded with a characteristic time of ~0.5 h (Supplementary file 2), which was ~2.5 fold faster than the corresponding TNB-labelled proteins, the observed characteristic times of fluorescence and FRET-monitored changes, when the DANS-labelled proteins were used as dopant proteins, were in qualitative agreement with those of the corresponding TNB-labelled proteins (Figure 1—figure supplement 2, Figure 4—figure supplement 3 and Supplementary file 2). The lack of correlation between the timescales of misfolding of the labelled proteins by themselves, and timescales when the same proteins were used as dopant proteins, suggests that the fast changes in FRET efficiency are unlikely to be a consequence of altered co-oligomerization kinetics of 98 μM Trp-less moPrP when it co-oligomerizes with 2 μM dopant.”

We also point out that in our previous analysis of the rise and fall FRET kinetics seen for the 197-169 and 197-223 segments, we had assumed that the conformational changes were happening irreversibly. Our new SEC experiments, described below, have shown that two different oligomer populations (O_S_ and O_L_) form reversibly from monomer M. We have re-analyzed our kinetic FRET data taking into account the SEC data, and we show that the two steps of FRET change seen for the 197-169 and 197-223 segments, occur during the formation of O_L_ from M and of O_S_ from O_L_. The analysis is described in the second paragraph of the subsection “Kinetic modelling of FRET and SEC data”, and also described below in response to a later point of the reviewers.

We also agree that the CD measurements are dominated by the oligomerization behavior of excess Trp-less moPrP. However, steady state tryptophan fluorescence anisotropy has no contribution from Trp-less moPrP, as it reports only on oligomers containing Trp. SS Trp fluorescence anisotropy therefore does not report on the oligomerization of Trp-less moPrP alone; it either reports solely on the oligomerization of 2 μM labeled moPrP, or the co-oligomerization of Trp-less and labeled moPrP. In the fourth paragraph of the subsection “Unlabelled and TNB-labelled mutant variants form misfolded co-oligomers with Trp-less moPrP with comparable kinetics”, we describe how our data clearly indicate that 2 μM labeled moPrP does not oligomerize independently when mixed with 98 μM Trp-less protein.

CD measurements at 10 or 20% dopant revealed a 50% higher global rate of misfolding when fit to a single exponential, which may in fact indicate faster initial association of TNB-labeled monomers even with WT PrP. Clear double exponential behavior may be hard to detect, given the limited resolution (Figure 2B) and the expected low amplitude of a fast phase when using excess WT PrP.

We agree that clear double exponential behavior may be hard to detect, given the limited resolution and the low amplitude of the fast phase. We have discussed this in the first paragraph of the subsection “Unlabelled and TNB-labelled mutant variants form misfolded co-oligomers with Trp-less moPrP with comparable kinetics” where we have stated: “TNB-labelled W197-C223 moPrP was chosen as the dopant as it misfolds nearly 10-fold faster by itself at 100% labelling density, compared to Trp-less moPrP (Supplementary file 2). […] It should be noted that although the observed kinetics appear to be described well by a single exponential equation, it is not possible to rule out the presence of two exponential components, with one component too small in amplitude to be detected.”

These concerns about mutant and/or label effects on the FRET measurements are reinforced by the presented model, proposing that the fast conformational change, i.e. helix compaction, occurs at a rate of 0.6-0.7 h-1 only AFTER oligomerization, yet all non-FRET readouts monitoring this oligomerization (e.g. anisotropy) show rates of ~ 0.1 h-1. How is that possible? Maybe the helix compaction and segment separation actually happen BEFORE oligomerization, in a spontaneous misfolding event that precedes assembly? This question relates to the essential debate about whether natively folded PrP fluctuates into a non-native structure before joining an aggregate, or whether it joins the aggregate and then re-configures.

We thank the reviewer for pointing out that the fast conformational change monitored by FRET may not be happening after the oligomer forms, as its rate constant is faster than the anisotropy monitored oligomerization rate constant.

On the other hand if the 0.6-0.7 h^-1^ FRET change were taking place in the monomer, before oligomerization, then it should be concentration independent, and would be observable even at the low concentration of 2 μM of labeled protein, in the absence of any Trp-less moPrP. However, we find that this conformational change is only observable at a total protein concentration of 100 μM, in the presence of 98 μM Trp-less moPrP.

We discuss this in the fourth paragraph of the subsection “Unlabelled and TNB-labelled mutant variants form misfolded co-oligomers with Trp-less moPrP with comparable kinetics” where we state: “The fluorescence spectra of the native monomers of the different unlabelled and labelled moPrP variants differed from those of the corresponding misfolded oligomers (Figure 1). […] These results also suggested that in the presence of 98 µM Trp-less moPrP, it was very unlikely that 2 µM dopant protein could misfold before oligomerization, and supported the results of the steady state Trp fluorescence anisotropy measurements which had indicated that the two proteins co-oligomerized.”

We describe, in our answer to the next point, how we have resolved the conundrum.

To draw more solid conclusions, the authors need to use a more direct technique (e.g. FCS, SEC, DLS, AFM and/or inter-molecular-FRET) for assessing the formation kinetics as well as size changes with time for the PrP variants, especially the Trp-less PrP. How do the amount and size of oligomers change with time? Measuring the diffusion constant (and hence size) by FCS as a function of time may allow the authors to assess whether the FRET-labelled molecules are changing conformation before or after becoming incorporated into the oligomer.Alternatively, the authors could consider applying some size fractionation method at different time-points to see whether labelled PrP has joined an oligomer before or after (or during) the fast structural change. This approach would have the advantage that different fractions could be characterized regarding their FRET efficiency, providing extra information about how the structure changes along with the growth of the oligomer.

To resolve the matter, and to reach more solid conclusions, it was necessary to carry out new experiments, as suggested by the reviewers. Also importantly, it was important for us to remember our own results on oligomerization monitored by size exclusion chromatography SEC, published over the past decade.

In our earlier studies of wt moPrP, as well as of different mutant moPrP variants, we had shown that the oligomerization reaction leads to the formation of two oligomer populations, the large O_L_ population and the smaller O_S_ population. It was shown that the transitions between monomer (M) and O_L_ and O_S_ are reversible, and that during oligomerization the populations of the two oligomers change with respect to each other. Also importantly, O_L_ could transform directly into O_S_.

We have now monitored oligomerization kinetics of 100 μM Trp-less moPrP using size-exclusion chromatography. In agreement with previous results with WT moPrP at pH 4, O_L_ and O_S_ were found to be populated. The populations of these oligomers changed, but their sizes (indicated by their elution volumes) remained fixed as oligomerization progressed. At the end of 60 h, we found the populations of monomer and oligomer (O_S_ and O_L)_to be ~14% and ~86% respectively (by integrating the area under each curve). Again in agreement with previous results, M, O_L_ and O_S_ were seen to be interconverting. We tested sequential, parallel and triangular kinetic models by carrying out kinetic simulations and global fitting, and found that the sequential model M↔O_L_↔O_s_ was able to account for both the size-exclusion and FRET data satisfactorily, with rate constants k_1,_ k_-1_, k_2_ and k_-2_ of 0.09, 0.05, 0.09 and 0.03 h^-1,^respectively.

In view of these new results and our previous results, it is clear that our earlier analysis, where (i) we did not incorporate the formation of two oligomers and (ii) assumed the conversion between the monomer and oligomer to be irreversible, was obviously incorrect. The incorrect analysis also meant that the rate constants that we had obtained assuming irreversibility and only one oligomer population were erroneous.

To globally fit the FRET data, the FRET efficiency values for M, O_L_ and O_S_ were allowed to vary locally for each sequence-segment. Unfortunately, the FRET efficiency values of O_L_ and O_S_ could not be determined experimentally as the SEC column was not able to separate them adequately. We note that the global fitting routine predicted a high FRET efficiency value for O_L_ and a low FRET efficiency value for O_S_ for the 197-169 and 197-223 sequence segments.

The new data and analysis therefore resolve the question of when during oligomerization, do the fast FRET changes take place. They indicate that compaction of the erstwhile helical segment takes place along the course of formation of O_L_ from M. The subsequent elongation of the erstwhile helical segments, takes place concomitant with the disassembly of O_L_ to form O_S_.

We discuss this in the subsection “Oligomer formation monitored by size-exclusion chromatography” where we state: “To directly monitor the change in population and/or size of the oligomers, size-exclusion chromatography (SEC) was carried out. In agreement with previous results obtained with WT moPrP (Singh et al., 2014), Trp-less moPrP was found to form two populations of oligomers, the larger O_L_ and the smaller O_S_, at pH 4 and 150 mM NaCl, As oligomerization progressed, the total amount of oligomers increased, but the sizes (as estimated from their elution volumes) remained fixed (Figure 3B). […] The poor resolution of the SEC column made it difficult to reliably estimate the relative amounts of O_L_ and O_S_ at equilibrium, but it appeared that, O_S_ was populated to about a three-fold higher extent than was O_L_.”

And we also state in the subsection “Kinetic modelling of FRET and SEC data”: “Kinetic simulations and global fitting were carried out to test whether a parallel, sequential or triangular model best described the formation of O_L_ and O_S_ from M, taking into account both the SEC and FRET data. […] It is more appropriate to use the M↔O_L_ ↔ O_S_ mechanism, because it is a simpler kinetic model than the triangular model.”

2) Another issue lies in the fact that mutating multiple Trp residues likely affects protein structure, stability, and dynamics (i.e. activation barriers), which raises questions about the robustness of the conclusions and their applicability. For instance, the Trp-less PrP shows the slowest kinetics, which may indicate that it has some deficiency in conformational conversion and oligomerization. As a minimum, the authors should use an additional method to verify whether the oligomer of Trp-less PrP has similar properties and formation efficiency as the WT oligomers. Whether the fluorescence/FRET measurement using TrP-less PrP in the dopant experiment really represent the kinetics and mechanism of the WT normal PrP is unclear. This issue is intrinsic to the approach and cannot be resolved for the current study, but should be discussed accordingly.

We have now shown using size-exclusion chromatography, that Trp-less moPrP forms both oligomers O_S_ and O_L_, to similar extents as compared to WT moPrP at pH 4 in the presence of 150 mM NaCl. We had characterized the oligomers of Trp-less moPrP (and the labelled and unlabeled mutant variant proteins) using DLS measurements and found them to be comparable to those of WT moPrP in an earlier paper. We have now included the DLS results for Trp-less and moPrP oligomers in this paper for better comparison. Furthermore, we have acknowledged the intrinsic caveat in this study in which Trp-less moPrP has been used as a pseudo WT moPrP analogue.

We discuss this in the Discussion as follows: “The presence of eight tryptophan residues in WT moPrP prevented its use in suppressing the intermolecular contributions to the FRET signal during oligomer formation. […] With this caveat in mind, the FRET measurements have allowed the delineation of the major structural changes that take place in the monomer, as it converts into soluble β-sheet rich oligomers at pH 4.”

An important aspect to consider for the W197 mutants, however, is that the equivalent residue in human PrP, F198, is the site of a pathogenic mutation. F198S is associated with GSS syndrome, an inherited prion disease. That the W197 mutants of moPrP show faster misfolding and oligomerization in the present study could therefore be related to the fact that misfolding is particularly sensitive to this residue, even though an F to W mutation is much more conservative than the F to S mutation responsible for GSS syndrome.

We note that for the mutant variants W197-C169 and W197-C223, only the former shows a ~2 fold increase in misfolding rates, whereas the latter remains unchanged, compared to WT moPrP (Supplementary file 2). Therefore, it cannot be concluded that the misfolding rate is faster in W197-C169 solely due to the F to W mutation. Instead, the faster misfolding rate for the W197-C169 mutant variant is possibly a result of the S169 to C169 mutation. This residue, located in the loop between β2 and α2 has been shown to be important in controlling the rigidity of the loop and affect the misfolding/aggregation propensity of the prion protein from different species.

We have discussed this as follows in the Supplementary Materials: “The unlabelled W197-C169 and W197-C223 mutant variants used here were not destabilized, supporting earlier observations that the F197W mutation does not perturb stability and/or unfolding/folding kinetics despite it being equivalent to residue F198 in human PrP, a site for the pathogenic mutation F198S associated with the inherited prion disease, the GSS syndrome (Jenkins et al., 2008). However, the W197-C169 mutant variant misfolded with slightly faster kinetics, compared to the WT moPrP, possibly due to its location in the loop between β2 and α2 (Agarwal et al., 2015; Gossert et al., 2005).”

3) There were initial concerns among the reviewers about the number of novel findings and the apparent overall advance provided by this work. Several previous studies had already concluded that the α2-α3 and β1-α1-β2 segments have to separate and α1 has to unfold prior to the β-conversion, leaving the potential compaction of the α2-α3 helices as the only major new result. However, these concern could be somewhat alleviated during the reviewers' discussions and a more in depth review of the literature, which revealed that previous work, for instance by Eghiaian et al. and Hafner-Bratkovic et al. using thermal denaturation or disulfide crosslinking, were less conclusive and more questionable regarding their relevance for the 'natural' misfolding process of PrP. In contrast, the present paper shows that domain separation occurs spontaneously in acidic conditions, which are endogenously found in endosomes. The authors should therefore better emphasize these advances of their study and in more detail put them into perspective to previous conclusions. Moreover, the somewhat turgid and disjointed writing of the paper does not do full justice to the assays and results. The manuscript is also strongly dominated by the setup of the system and various controls, whereas critical experiments and their discussion are proportionally slim, which should be improved in a revised version.

We thank the reviewers for pointing out that we need to better present the importance of our findings, especially in the context of earlier work.

We now state in the fifth paragraph of the Introduction: “While such equilibrium studies have suggested possible sequences of structural changes during oligomer formation (Singh and Udgaonkar, 2016b) there has been a dire need for kinetic studies that can directly delineate the structural mechanism of the misfolding which accompanies the oligomerization of the prion protein at low pH.”

We also now state in the fourth paragraph of the Discussion: “A qualitative comparison of the characteristic times of all structural changes monitored by FRET suggests that a compaction of the segments spanning the α2 and α3 helices is the fastest change (Table 1). […] The slowest change appears to be the elongation of sequence segments that had spanned α2 and α3; this is possibly due to their conversion into β-sheets and is also reported on by the global probe CD (Figure 6).”

In the seventh paragraph of the Discussion, we now describe our results in the context of previous hydrogen exchange-mass spectrometry measurements.

By carrying out SEC measurement of the oligomerization process, and by correlating the results of these measurements with the FRET measurements, we have now been able to describe a kinetic model for the formation of the two types of oligomers that are seen to form. Importantly we have been able to delineate the structural changes that occur during each of the two steps of oligomer formation. This is described in detail in the aforementioned paragraph.

In addition, we have now shortened our description of various controls and focussed more on the discussion of the critical results only.

We believe that the importance of our results is much better brought out in the revised manuscript.